# Neural Models for Output-Space Invariance in Combinatorial Problems

**Yatin Nandwani,**[*] **Vidit Jain,**[*] **Mausam & Parag Singla**
Department of Computer Science, Indian Institute of Technology Delhi, INDIA
{yatin.nandwani, vidit.jain.cs117, mausam, parags}@cse.iitd.ac.in

## Abstract

Recently many neural models have been proposed to solve combinatorial puzzles by implicitly learning underlying constraints using their solved instances, such as sudoku or graph coloring (GCP). One drawback of the proposed architectures, which are often based on Graph Neural Networks (GNN) (Zhou et al., 2020), is that they cannot generalize across the size of the output space from which variables are assigned a value, for example, set of colors in a GCP, or board-size in sudoku. We call the output space for the variables as 'value-set'. While many works have demonstrated generalization of GNNs across graph size, there has been no study on how to design a GNN for achieving value-set invariance for problems that come from the same domain. For example, learning to solve $16 \times 16$ sudoku after being trained on only $9 \times 9$ sudokus, or coloring a 7 colorable graph after training on 4 colorable graphs. In this work, we propose novel methods to extend GNN based architectures to achieve value-set invariance. Specifically, our model builds on recently proposed Recurrent Relational Networks (RRN) (Palm et al., 2018). Our first approach exploits the graph-size invariance of GNNs by converting a multi-class node classification problem into a binary node classification problem. Our second approach works directly with multiple classes by adding multiple nodes corresponding to the values in the value-set, and then connecting variable nodes to value nodes depending on the problem initialization. Our experimental evaluation on three different combinatorial problems demonstrates that both our models perform well on our novel problem, compared to a generic neural reasoner. Between two of our models, we observe an inherent trade-off: while the binarized model gives better performance when trained on smaller value-sets, multi-valued model is much more memory efficient, resulting in improved performance when trained on larger value-sets, where binarized model fails to train.

## 1 Introduction

The capability of neural models to perform symbolic reasoning is often seen as a step towards the framework for unified AI, *i.e.*, building end-to-end trainable system for tasks, which need to combine low level perception with high level cognitive reasoning (Kahneman, 2011). While neural networks are naturally excellent at perception, they are increasingly being developed for high-level reasoning tasks, *e.g.*, solving SAT (Selsam et al., 2019; Amizadeh et al., 2019a;b), neural theorem proving (Rocktäschel et al., 2015), differentiable ILP ($\partial$ILP) (Evans & Grefenstette, 2018), playing blocks world (Dong et al., 2019), solving sudoku (Wang et al., 2019). Our work follows this literature for solving combinatorial puzzles – in particular, the methods that implicitly incorporate the rules in their weights by training over some of its solved instances, *e.g.* Recurrent Relational Networks (RRN) (Palm et al., 2018). Such models assume a fixed value-set, *i.e.*, the set from which variables are assigned values is assumed to be constant during training and testing. This is a significant limitation, since it may not always be possible to generate sufficient training data for similar large problems in which variables take values from a bigger value-set (Najafian et al., 2018). It is also a desirable goal since as humans, we often find it natural to generalize to problems of unseen variable and value sizes, once we know how to solve similar problems of a different size, *e.g.*, we may solve a $12 \times 12$

---

[*]Equal contribution. Work done while at IIT Delhi. Current email: vidit.jain@alumni.iitd.ac.in

sudoku after learning to solve a $9 \times 9$ sudoku. We note that graph based models have been shown to generalize well on varying graph sizes, *e.g.*, finding a satisfying solution of a CNF encoding of a CSP with 100 Boolean-variables, after training on CNF encodings of CSPs with only 40 Boolean-variables (Selsam et al., 2019). However, the model trained using CNF encoding of Boolean-CSPs cannot be used directly for a non-Boolean CSP in which variables take value from a different (larger) value-set.

In response, we study value-set invariance in combinatorial puzzles from the same domain. To formally define a similar puzzle with variables taking values from a different value-set, we make use of *Lifted CSP* (Joslin & Roy, 1997), a (finite) first-order representation that can be ground to CSPs of varying variable and value-set sizes. We note that even though we use Lifted CSPs to define value-set invariance, its complete specification is assumed to be unknown. Specifically, we do not have access to the constraints of the CSP, and thus neural SAT solvers like NeuroSAT (Selsam et al., 2019) can not be used. While training, we only assume access to solved instances along with their constraint graph. We define our problem as: given solved instances and corresponding constraint graph of an unknown ground CSP with a value-set of size $k$, can we learn neural models that generalize to instances of the same lifted CSP, but with a different value-set of size $k'$ (typically $k' > k$)? An example task includes training a model using data of $9 \times 9$ Sudoku, but testing on a $12 \times 12$ or a $16 \times 16$ Sudoku. We build our solution using RRNs as the base architecture. They run GNN on the constraint graph, and employ iterative message passing in a recurrent fashion – the nodes (variables) are then decoded to obtain a solution. We present two ways to enhance RRNs for value-set invariance.

*Binarized Model:* Our first model converts a multi-class classification problem into a binary classification problem by converting a multi-valued variable into multiple Boolean variables, one for each value in the value-set. The binarized constraint graph gets defined as: if there is an edge between two variables in original constraint graph, there are $k$ edges between Boolean nodes corresponding to the same value and the same two variables in the new graph. In addition, all $k$ Boolean variables, corresponding to a multi-valued variable, are connected with each other. This model naturally achieves value-set invariance. At test time, a larger value-set just results in a larger graph size. All GNN weights are tied, and because all the variables in the binarized model are Boolean, embeddings for binary values '0' and '1', trained at training time, are directly applicable at test time.

*Multi-valued Model:* Our second model directly operates on the given multi-valued variables and the corresponding constraint graph, but introduces a *value node* for every value in the value-set. Each pre-assigned (unassigned) variable node is connected to that (respectively, every possible) value node. The challenge in this model is initializing value nodes at test time when $k' > k$. We circumvent this problem by training upfront $k'$ or more value embeddings by randomly sub-selecting a $k$ sized subset during each learning iteration. This random sub-selection exploits the symmetry of value-set elements across instances. During test time, $k'$ of the learned embeddings are used.

We perform extensive experimental evaluation on puzzles generated from three different structured CSPs: Graph Coloring (GCP), Futoshiki, and Sudoku. We compare two of our models with an NLM (Dong et al., 2019) baseline – a generic neural reasoner, which either fails to scale or performs significantly worse for most test sizes used in our experiments. We also compare our two models along the axes of performance and scalability and discuss their strengths and weaknesses.

## 2 RELATED WORK

This paper belongs to the broad research area of neural reasoning models, in which neural models learn to solve pure reasoning tasks in a data-driven fashion. Some example tasks include theorem proving (Rocktäschel et al., 2015; Evans & Grefenstette, 2018), logical reasoning (Cingillioglu & Russo, 2019), probabilistic logic reasoning (Manhaeve et al., 2018), classical planning (Dong et al., 2019), probabilistic planning in a known MDP (Tamar et al., 2017; Bajpai et al., 2018), and our focus – combinatorial problems that are instances of an unknown constraint satisfaction problem.

There are two main research threads within neural CSPs and SAT. First thread builds neural models for problems where the CSP constraints or SAT clauses are *explicitly* provided to the model. For example, NeuroSAT (Selsam et al., 2019) and PDP (Amizadeh et al., 2019b) assume that the CSP is expressed in a Conjunctive (or Disjunctive) Normal Form. Similarly, Circuit-SAT (Amizadeh et al., 2019a) uses the knowledge of exact constraints to convert a CSP into a Boolean Circuit. This research has similarities with logical reasoning models like DeepProbLog (Manhaeve et al., 2018),

and DeepLogic (Cingillioglu & Russo, 2019), which require human designed rules for reasoning. Our work belongs to the second thread where the constraints or clauses are not provided explicitly, and only some underlying structure (e.g., Sudoku grid cell connectivity) is given along with training data. The intention is that the model not only learns to reason for the task, but also needs to learn the implicit semantics of each constraint. SATNET (Wang et al., 2019) falls in this category – it formulates a learnable low-rank Semi-definite Program (SDP) relaxation for a given MAXSAT problem trained via solved SAT problems. Similarly, Recurrent Relational Networks (RRN) (Palm et al., 2018) use recurrent message passing graph neural network to embed the variables of the unknown CSP, and the relationship between them, in a latent vector space and finally assign a value to each variable based on its embedding. Both these works assume a fixed number of variables that remains unchanged across training and test. While we build on RRNs, we substantially extend the formalism to study value-set invariance. Formally, our work can be seen as solving a (finite) first-order formulation of the CSP, called *Lifted* CSP (Joslin & Roy, 1997), which can be grounded to CSPs with varying number of variables and values. To our knowledge, there is relatively limited prior work on neural models that can generalize to variable-sized instances of an underlying first order reasoning task – one related approach builds neural models for First-order MDPs (Garg et al., 2020).

Finally, there has been a long history of work dedicated to learning rules or constraints from training data using Inductive Logic Programming (Lavrac & Raedt, 1995; Friedman et al., 1999). Evans & Grefenstette (2018) propose differentiable neural relaxation of ILP ($\partial$ILP). Neural Logic Machines (NLM) (Dong et al., 2019) is another framework that learns lifted rules, shown to be more scalable than $\partial$ILP. It allows learning of first-order logic rules expressed as Horn Clauses over a set of predicates. Learning of first-order rules makes NLM amenable to transfer over different CSP sizes (Nandwani et al., 2021), and are thus directly comparable to our work. The main challenge of such approaches is that they fail to scale to the size of the problems considered in this work. In our experiments, we compare our methods against both deep and shallow versions of NLM. Note that our work relies on the assumption that GNNs generalize across graph sizes. Yehudai et al. (2021) study the scenarios under which this assumption may not hold. We discuss the details in the appendix.

## 3 PRELIMINARIES AND PROBLEM DEFINITION

A combinatorial puzzle can be thought of as a grounded CSP and to formally define a puzzle from the same domain but a larger value-set, we resort to the notion of '*Lifted CSPs*' that represent an abstraction over multiple ground CSPs of the same type. A lifted CSP does not include a specific set of variables and values; instead, it operates in terms of variable and value *references* that can be instantiated with all ground variables and values in a ground CSP. This makes them amenable to instantiate CSPs or puzzles with varying number of variables as well as values. We define a Lifted CSP $\mathcal{L}_C$ as a three tuple $\langle \mathcal{P}, \mathcal{R}, \mathcal{C} \rangle$. $\mathcal{P}$ is a set of predicates: a predicate $p \in \mathcal{P}$ represents a Boolean function from the set of its arguments, which are variable references. Similarly, $\mathcal{R}$ is a set of relations over value space – a $r \in \mathcal{R}$ reprents a Boolean function over arguments that are value references. A predicate (or a relation) with its arguments is called an atom. $\mathcal{C}$ is a set of lifted constraints, constructed by applying logical operators to atoms – they are interpreted as universally quantified over all instantiations of variable and value references. Finally, Lifted CSP uses a special unary function `Value`, whose argument is a variable reference and evaluates to a value reference. As an example, a lifted CSP for Sudoku may have a $\mathcal{P} = \{$`Nbr`$\}$ for whether two cells are in same row, column or box, $\mathcal{R} = \{$`Neq`$\}$, representing two values are unequal, and a lifted constraint: `Nbr`$(c_1, c_2) \rightarrow$ `Neq`(`Value`$(c_1)$, `Value`$(c_2)$).

A lifted CSP $\mathcal{L}_C$ yields a ground CSP $C$, given a set of variables $\mathcal{O}$, and a set of values $\mathcal{V}$, and a complete instantiation of all predicates and relations over this set (e.g., in Sudoku, the number of cells, possible values, and which cells are neighbors and which are not). The ground constraints are constructed by instantiating lifted constraints over all variables and values. A (satisfying) solution, $\mathbf{y}$, of a CSP refers to a complete specification of `Value`: $\mathcal{O} \rightarrow \mathcal{V}$ function, such that all the constraints are satisfied. We are often given a partial (satisfying) solution, $\mathbf{x}$ – an assignment of values to a subset of variables $\tilde{\mathcal{O}} \subseteq \mathcal{O}$ and the goal is to output $\mathbf{y}$, such that $\mathbf{y}$ agrees with $\mathbf{x}$ for the subset $\tilde{\mathcal{O}}$.

Given a ground CSP $C$, the *Constraint Graph*, $G_C = (N_C, E_C)$, is constructed by having each variable in the CSP represent a node in the graph and introducing an edge between two nodes $n_1^C, n_2^C$ iff the corresponding variables appear together in some constraint. The edges in the constraint graph

149 are typed based on the identity of the lifted constraint from which it comes. Note that there could
150 be multiple edges between nodes $n_1^C, n_2^C$ in $G_C$, if these nodes appear together in more than one
151 constraint. We embed the knowledge about relations between values in $\mathcal{V}$ in the form of another
152 graph, called *Relation Graph*, $G_R = (N_R, E_R)$, where there is a node for every value in the set $\mathcal{V}$,
153 and there is a (directed) edge between nodes corresponding to $v_l, v_l'$ depending on whether $r(v_l, v_{l'})$
154 is true or not, for every $r \in \mathcal{R}$. Similar to $G_C$, this graph can also have multi-edges between two
155 pairs of nodes, if more than one relationship holds between the corresponding values.

**Problem Definition:** To achieve value-set invariance, our goal is to train a model $M_\Theta$ on training
157 data from an unknown ground CSP $C$ (with variables $\mathcal{O}$ and value-set $\mathcal{V}$) obtained from an unknown
158 lifted CSP $\mathcal{L}_C$, and test it on an arbitrary ground CSP $C'$ from the same lifted CSP (with variables
159 $\mathcal{O}'$ and value-set $\mathcal{V}'$), where $|\mathcal{V}| \neq |\mathcal{V}'|$. Formally, we are given training data $\mathcal{D}$ as a set of tuples
160 $\{((\mathbf{x^i}, G_{C^i}), \mathbf{y^i})\}_{\mathbf{i}=1}^M$, along with a relationship graph $G_R$ encoding relations between values in the
161 value-set $\mathcal{V}$. Here, $\mathbf{i}^{th}$ instance denotes a partial and corresponding complete solution for $C^\mathbf{i}$. We note
162 that explicit form of the constraints in $C^\mathbf{i}$ or $\mathcal{L}_C$ are not available, only the graphs are given to the
163 model. Our goal is to learn model $M_\Theta$, such that given graphs $G_{C'}$ and $G_{R'}$, and a partial solution
164 $\mathbf{x}'$ (for CSP $C'$) : $M_\Theta(\mathbf{x}') = \mathbf{y}'$, only if $\mathbf{y}'$ is a corresponding complete solution for $\mathbf{x}'$. Note that in
165 one of our models, we will additionally assume that $\max |\mathcal{V}'|$, denoted as $k_{\max}$, is known to us at
166 training time, which we argue is a benign assumption for most practical applications.

# 4 Models Description

168 We propose two models for value-set invariance: the
169 *Binarized Model*, and the *Multi-valued Model*. In
170 each case, we assume the training data is provided in
171 the form $\mathcal{D} = (\{(\mathbf{x^i}, G_{C^i}), \mathbf{y^i}\}_{\mathbf{i}=1}^M, G_R)$ as described
172 in Section 3. Let $\mathcal{V}$ and $\mathcal{V}'$ denote the value-sets at
173 train and test time, with cardinality $k, k'$, respectively.
174 For each model, we first present a high level intu-
175 ition, followed by description of: (a) Construction of
176 Message Passing Graph (b) Message Passing Rules
177 (c) Loss Computation, and finally (d) Prediction on
178 a problem with larger value-set.

## 4.1 Binarized Model

180 Intuition behind our Binarized Model comes directly
181 from the 'sparse encoding' of a discrete CSP into a
182 SAT formula (de Kleer, 1989; Walsh, 2000), in which
183 assignment of a value $v \in \mathcal{V}$ to any variable $\mathbf{x[j]} \in$
184 $\mathcal{O}$ is encoded by a Boolean variable that represents
185 $\mathbf{x[j]} == v$. Such an encoding converts a single multi-

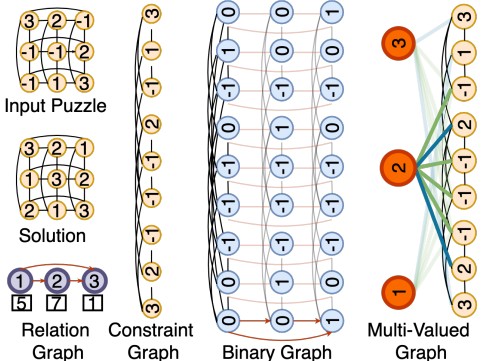

Figure 1: An example Futoshiki Puzzle of size $3 \times 3$ and the corresponding graphs. A value of $-1$ indicates an unassigned variable. Black and red edges are Constraint and Relation edges respectively. The digits $5, 7, 1$ in square boxes represent a random 3-permutation of $k_{\max}$, used in multi-valued model for initialization of node embeddings.

186 valued variable into multiple Boolean valued variables.[1] We convert a Constraint Graph (fig. 1)
187 with nodes representing multi-valued variables (yellow nodes), into a Binary Graph (fig. 1) with
188 Boolean nodes (blue nodes). This creates a $|N_C| \times k$ grid of Boolean nodes, with a row representing
189 a variable, a column representing a value and a grid cell (a Boolean node) representing assignment of
190 a particular value to a particular variable. Such a graph can easily represent relationship between the
191 values as well (horizontal red edges), thereby encapsulating the information present in the Relation
192 Graph (fig. 1). We use this Binary Graph for message passing.

**Construction of Message Passing Graph:** We denote the *Message Passing Graph (MPG)* by
194 $G = (N, E)$ with the set of nodes $N$ and set of edges $E$, constructed as follows: **Nodes:** For each
195 node $n_\mathbf{j}^C \in N_C$ in the Constraint Graph (fig. 1, yellow nodes), we construct $k$ binary valued nodes,
196 denoted as $n_{\mathbf{j},1}, n_{\mathbf{j},2} \cdots n_{\mathbf{j},k}$ in $N$ (blue nodes in Binary Graph). **Edges:** We construct two categories
197 of edges in $G$. The first category of edges are directly inherited from the edges of the constraint
198 graph $G_C$ (black vertical edges), with $k$ copies created due to binarization. Edge type is same as in
199 the original constraint graph and is denoted by $q$. Formally, for every edge, $e^C_{(\mathbf{j},\mathbf{j}')} \in E_C$, where

---

[1]There is an alternative encoding scheme called 'compact encoding'. It is discussed in the appendix

$e^C{}_{(\mathbf{j},\mathbf{j}')}.type = q$, we introduce $k$ edges denoted as $e^q_{(\mathbf{jl},\mathbf{j'l})}$, i.e., there is an edge between every pair of nodes, $n_{\mathbf{j},\mathbf{l}}$ and $n_{\mathbf{j'},\mathbf{l}}$, $1 \leq \mathbf{l} \leq k$. We refer to them as *Constraint Edges*. The second category of edges encode the information from the Relationship Graph $G_R$ into the MPG, with $|N_C|$ copies of it created, one for each variable. For every edge $e^R{}_{(\mathbf{l},\mathbf{l}')} \in E_R$ with edge type $r$, create an edge $e^r_{(\mathbf{jl},\mathbf{jl}')}$ with type $r$ between every pair of binary nodes $n_{\mathbf{j},\mathbf{l}}$ and $n_{\mathbf{j},\mathbf{l}'}$, $1 \leq \mathbf{j} \leq |N_C|$ (*e.g.*, red edges encoding *less-than* relation between value pairs $(1, 2)$, $(2, 3)$ and $(1, 3)$). We refer to them as *Relational Edges*.

**Recurrent Message Passing:** Once MPG has been constructed, we follow recurrent message passing rules, with weights shared across layers, similar to RRNs (Palm et al., 2018) with some differences. For each node $n_{\mathbf{j},\mathbf{l}}$ in the graph, we maintain a hidden state $h_t(n_{\mathbf{j},\mathbf{l}})$, which is updated at each step $t$ based on the messages received from its neighbors. This hidden state is used to compute the probability of a binary node taking a value of 1. Since we use sparse encoding, only the node with maximum probability amongst the $k$ binary nodes $n_{\mathbf{j},\mathbf{l}}$; $1 \leq \mathbf{l} \leq k$, corresponding to multi-valued variable $\mathbf{x}[\mathbf{j}]$, is assigned a value 1, at the end of message passing. We give the details of message passing and state update function in appendix. Next, we discuss how the nodes are initialized before message passing starts, followed by the details of loss computation.

**Initialization:** Irrespective of the size of value-set $\mathcal{V}$ or vertices $N_C$, there are 3 learnable embeddings ($u[0]$, $u[1]$ and $u[-1]$) for initialization: two for binary values 0 and 1, and one for value $-1$ representing unassigned nodes. All $k$ nodes corresponding to an unassigned variable $\mathbf{x}[\mathbf{j}]$ are initialized with $u[-1]$, *i.e.*, whenever $\mathbf{x}[\mathbf{j}]$ is NULL (yellow nodes with $-1$), $u_0(n_{\mathbf{j},\mathbf{l}}) = u[-1], \forall v_{\mathbf{l}} \in \mathcal{V}$, where $u_0$ represents initial embedding function. On the other hand, if $\mathbf{x}[\mathbf{j}]$ is preassigned a value $v_{\hat{\mathbf{i}}}$, then $u_0(n_{\mathbf{j},\mathbf{l}}) = u[0], \forall v_{\mathbf{l}} \neq v_{\hat{\mathbf{i}}}$, and $u_0(n_{\mathbf{j},\hat{\mathbf{i}}}) = u[1]$. *E.g.*, variable corresponding to the binary nodes in 1st row has a preassigned value of '3', consequently, binary nodes in 1st and 2nd column of the 1st row are initialized with $u[0]$, and binary node in the 3rd column of 1st row, which corresponds to assignment '$\mathbf{x}[1] = 3$', is initialized with $u[1]$. Lastly, the hidden state, $h_0(n_{\mathbf{j},\mathbf{l}})$, of each node, $n_{\mathbf{j},\mathbf{l}}$, is initialized as a $\mathbf{0}$ vector, $\forall \mathbf{j}, \forall v_{\mathbf{l}}$.

**Loss Computation:** The Binary Cross Entropy (BCE) loss for each node $n_{\mathbf{j},\mathbf{l}}$ is computed w.r.t. its target, $\tilde{\mathbf{y}}[\mathbf{j}, \mathbf{l}]$, which is defined as 1 whenever $\mathbf{y}[\mathbf{j}] = \mathbf{l}$ and 0 otherwise. At each step $t \in \{1 \ldots T\}$, we can compute the probability $Pr(n_{\mathbf{j},\mathbf{l}}.v = 1; \Theta)$ of classifying a node $n_{\mathbf{j},\mathbf{l}}$ as 1 by passing its hidden state through a learnable scoring function $s$, *i.e.*, $Pr_t(n_{\mathbf{j},\mathbf{l}}.v = 1; \Theta) = \sigma(s(h_t(n_{\mathbf{j},\mathbf{l}})))$, where $\sigma$ is the standard Sigmoid function. Here, $n_{\mathbf{j},\mathbf{l}}.v$ denotes the value that node $n_{\mathbf{j},\mathbf{l}}$ can take and belongs to the set $\{0, 1\}$. Loss at step $t$ is the average BCE loss across all the nodes: $\frac{1}{|N|} \sum_{n_{\mathbf{j},\mathbf{l}} \in N} \tilde{\mathbf{y}}[\mathbf{j}, \mathbf{l}] \log Pr_t(n_{\mathbf{j},\mathbf{l}}.v = 1; \Theta) + (1 - \tilde{\mathbf{y}}[\mathbf{j}, \mathbf{l}]) \log Pr_t(n_{\mathbf{j},\mathbf{l}}.v = 0; \Theta)$. Like Palm et al. (2018), we back-propagate through the loss at every step $t \in \{1 \ldots T\}$ as it helps in learning a convergent message passing algorithm. During training, the objective is to learn the 3 initial embeddings $u[-1], u[0], u[1]$, functions used in message passing and state update, and the scoring function $s$.

**Prediction on a problem with larger size of value-set:** While testing, let the constraint and relation graph be $G_{C'}$ and $G_{R'}$ with $n'$ and $k'$ nodes respectively. Let $\mathbf{x}'$ be a partial solution, with $n'$ variables $\mathbf{x}'[\mathbf{j}]$, each taking a value from value-set $\mathcal{V}'$ of size $k'$. As described above, we create a graph $G'$ with $n'k'$ nodes, run message passing for $T$ steps, and for each variable $\mathbf{x}'[\mathbf{j}]$, compute the $k'$ probabilities, one for each of the $k'$ nodes $n_{\mathbf{j},\mathbf{l}} \forall \mathbf{l} \in \mathcal{V}'$ corresponding to the variable $\mathbf{x}'[\mathbf{j}]$, which is assigned the value corresponding to maximum probability, *i.e.*, $\hat{\mathbf{y}}[\mathbf{j}] = \arg\max_{\mathbf{l} \in \mathcal{V}'} Pr_T(n_{\mathbf{j},\mathbf{l}}.v = 1; \Theta)$.

## 4.2 Multi-valued Model

Multi-valued model differs from the binarized model by avoiding binarization of nodes, and instead explicitly adding *Value Nodes* in the message passing graph, one for each value in the value-set. The message graph consists of two components: (a) A Graph $G = (N, E)$ to represent constraints inherited from the constraint graph $G_C = (N_C, E_C)$ (b) A Graph $\tilde{G} = (\tilde{N}, \tilde{E})$ to represent relations inherited from the relationship graph $G_R = (N_R, E_R)$. We refer to $G$ as *Constraint Message Passing Graph* (**CMPG**), and $\tilde{G}$ as *Relationship Message Passing Graph* (**RMPG**). Message passing on RMPG first generates desired number of embeddings (upto $k_{\max}$), one for each of the value nodes. This is followed by message passing on CMPG which uses the embeddings of the value nodes generated by RMPG and computes embeddings for each variable node. Finally, the variable nodes are classified based on the similarity of their embedding with the embeddings of the value nodes

252 computed by RMPG. Learning to generate upto $k_{\max}$ embeddings from training samples with only
253 $k(< k_{\max})$ values in the value-set is the main technical challenge that we address in this model.

254 **Construction of CMPG:** **Nodes:** For each node $n_{\mathbf{j}}^C \in N_C$ in the constraint graph, we construct
255 a $k$-valued node, denoted as $n_{\mathbf{j}} \in N$. Total number of such nodes constructed is $|N_C|$. We refer
256 to these as *Variable Nodes* (yellow nodes in Multi-Valued Graph in fig. 1). Additionally, for each
257 value $v_{\mathbf{l}} \in \mathcal{V}$ in the value-set, we create a node, denoted as $n_{\mathbf{l}}^v \in N$. Total number of such nodes
258 constructed is $|\mathcal{V}|$. We refer to these as *Value Nodes* (orange nodes). **Edges:** For every edge,
259 $e^C_{(\mathbf{j},\mathbf{j}')} \in E_C$, where $e^C_{(\mathbf{j},\mathbf{j}')}.type = q$, we introduce an edge denoted as $e^q_{(\mathbf{j},\mathbf{j}')}$ with type $q$. These
260 edges are directly inherited from the constraint graph. We refer to these as *Constraint Edges* (black
261 edges). Additionally, to indicate the pre-assignment of values to the variables in $\mathbf{x}$, we introduce new
262 edges connecting value nodes to appropriate variable nodes. Whenever $\mathbf{x}[\mathbf{j}] = v_{\mathbf{l}}$, add an edge, $e^a_{(\mathbf{j},\mathbf{l})}$
263 between variable node $n_{\mathbf{j}}$ and value node $n_{\mathbf{l}}^v$ (blue edges). If $\mathbf{x}[\mathbf{j}]$ is NULL, *i.e.*, unassigned, then add
264 $k$ edges, $e^{\bar{a}}_{(\mathbf{j},\mathbf{l})}, \forall v_{\mathbf{l}} \in \mathcal{V}$, connecting the variable node $n_j$ with all $k$ value nodes $n_{\mathbf{l}}^v$ (*e.g.*, green edges
265 connecting orange value node '2' to all '-1' variable nodes). We refer to them as *Assignment Edges*.

266 **Construction of RMPG: Nodes:** For each value $v_{\mathbf{l}} \in \mathcal{V}$, create a node denoted as $\tilde{n}_{\mathbf{l}}^v \in \tilde{N}$ (purple
267 nodes in Relation Graph in fig. 1). Total number of such nodes constructed is $|\mathcal{V}|$. We refer to these
268 as *Value Nodes*. **Edges:** For every pair of value nodes, $\tilde{n}_{\mathbf{l}}^v$ and $\tilde{n}_{\mathbf{l}'}^v$, introduce an edge $\tilde{e}^r_{(\mathbf{l},\mathbf{l}')}$ with type
269 $r$ if $r(v_{\mathbf{l}}, v_{\mathbf{l}'})$ holds based on the relationship graph $G_R$, i.e., $e^R_{(\mathbf{l},\mathbf{l}')} \in E_R$ with edge label $r$ (red
270 edges). These edges are defined for relations that exist between values in the value-set.

271 **Achieving Value-set Invariance:** A key question arises here: why do we need to construct a separate
272 RMPG ($\tilde{G}$)? Why not embed relevant edges in CMPG ($G$), as done for the binarized model? The
273 answer lies in realizing that we represent each value in the value-set explicitly in the multi-valued
274 model, unlike the binarized model. Hence, our model needs to learn representation for each of them
275 in the form of value node embeddings. Further, to generalize we need to learn as many embeddings
276 as there are values in the largest test value-set, i.e., $k_{\max} = \max |\mathcal{V}'|$. We achieve this by randomly
277 sub-selecting a $k$-sized set from $\{1 \ldots k_{\max}\}$ and permuting the chosen subset for each training
278 example in a given mini-batch, and then computing the 'relationship-aware' embeddings from this
279 permuted subset through message passing in RMPG. The 'relationship-aware' embeddings are then
280 used to initialize the value nodes (orange nodes) during message passing in CMPG. For instance,
281 if the permutation obtained is $\{w_1, \cdots, w_{\mathbf{l}}, \cdots, w_k\}$, where $\forall \mathbf{l}, 1 \leq w_{\mathbf{l}} \leq k_{\max}$, then embedding
282 for the value node $\tilde{n}_{\mathbf{l}}^v$ in $\tilde{G}$ is initialized by $w_{\mathbf{l}}^{th}$ learnable embedding (*e.g.*, purple nodes for values
283 '1', '2', and '3' are initialized by the 5th, 7th, and 1st learnable embedding, respectively). After
284 message passing on $\tilde{G}$, the 'relationship-aware' embedding of $\tilde{n}_{\mathbf{l}}^v$ (purple node) is used to initialize
285 the embedding for value node $n_{\mathbf{l}}^v$ (orange node) in $G$. This elegant process is able to train all the $k_{\max}$
286 embeddings by simply using the training data corresponding to $\mathcal{V}$, and the corresponding relationship
287 information. Since these relationship aware embeddings need to be pre-computed before they can be
288 passed to the downstream constraint processing, we construct two different message passing graphs,
289 one for computing relationship-aware embeddings and one for constraint handling.

290 **Recurrent Message Passing on RMPG:** Rules of message passing and hidden state updates at
291 every step $t$ are similar to RRN in Palm et al. (2018) and defined in detail in the appendix. After
292 updating the hidden states for total $\tilde{T}$ steps, the final embeddings, $\tilde{h}_{\tilde{T}}(\tilde{n}_{\mathbf{l}}^v) \,\forall v_{\mathbf{l}} \in \mathcal{V}$, are used as
293 'relationship-aware' embeddings for initializing the input features (embeddings) of the nodes in
294 CMPG $G$. We now discuss the initialization of the value nodes before message passing in RMPG.

295 **Initialization:** There are a total of $k_{\max}$ learnable embeddings, $\tilde{u}[\mathbf{l}'], 1 \leq \mathbf{l}' \leq k_{\max}$, out of which any
296 $k$ are randomly chosen for initializing the nodes in RMPG. *e.g.*, $\tilde{u}[5], \tilde{u}[7], \tilde{u}[1]$ are chosen to initialize
297 the purple value nodes '1','2', and '3' in Relation Graph in fig. 1. Formally, for each input $\mathbf{x}$, select
298 a $k$-permutation, $\mathcal{P}_{\mathbf{x}}$, of $k_{\max}$. Initialize the embedding of $\tilde{n}_{\mathbf{l}}^v$ in $\tilde{G}$ with $\tilde{u}[\mathcal{P}_{\mathbf{x}}[\mathbf{l}]], \,\forall \mathbf{l} \in \{1 \ldots k\}$.
299 Initialize the hidden state, $\tilde{h}_0(\tilde{n}_{\mathbf{l}}^v), \,\forall \tilde{n}_{\mathbf{l}}^v \in \tilde{N}$ with a $\mathbf{0}$ vector.

300 **Recurrent Message Passing on CMPG:** Message passing on CMPG updates the hidden state,
301 $h_t(n_{\mathbf{j}})$, of each variable node $n_{\mathbf{j}}$ for a total of $T \, (t \leq T)$ steps using the messages received from its
302 neighbors. The details are similar to message passing in binarized model and are discussed in the
303 appendix. Below we describe the initialization of node embeddings followed by computation of loss.

Table 1: Futoshiki: Mean (Std. dev.) of Board and Pointwise accuracy on different board sizes. MV and BIN correspond to Multi-valued Model and Binarized Model, respectively.

| | 6 | 7 | 8 | 9 | 10 | 11 | 12 |
|---|---|---|---|---|---|---|---|
| **Board Accuracy** | | | | | | | |
| NLM15 | 73.37 (1.34) | 56.98 (1.47) | 48.71 (1.96) | 44.16 (1.72) | 37.54 (2.74) | 32.50 (2.84) | - |
| NLM30 | 85.72 (0.39) | 69.61 (0.57) | 63.52 (1.20) | 60.73 (1.29) | 55.94 (0.85) | - | - |
| MV | 99.62 (0.18) | 90.18 (2.38) | 71.58 (4.66) | 54.85 (6.89) | 38.51 (5.62) | 24.18 (4.49) | 11.97 (5.54) |
| BIN | **99.86 (0.01)** | **97.92 (1.27)** | **93.39 (4.08)** | **89.39 (6.03)** | **83.48 (10.7)** | **76.14 (15.83)** | **68.15 (22.08)** |
| **Pointwise Accuracy** | | | | | | | |
| NLM15 | 96.72 (0.16) | 93.9 (0.26) | 93.43 (0.26) | 93.86 (0.28) | 94.07 (0.29) | 94.29 (0.31) | - |
| NLM30 | 97.88 (0.05) | 95.32 (0.10) | 95.09 (0.14) | 95.48 (0.08) | 95.68 (0.03) | - | - |
| MV | 99.91 (0.03) | 98.84 (0.24) | 97.09 (0.46) | 96.07 (0.60) | 95.17 (0.53) | 94.52 (0.41) | 93.99 (0.60) |
| BIN | **99.97 (0.00)** | **99.63 (0.13)** | **99.02 (0.37)** | **98.60 (0.47)** | **98.23 (0.68)** | **97.85 (0.98)** | **97.66 (1.31)** |

**Initialization:** We initialize the embedding of value nodes (orange nodes), $n_{\mathbf{l}}^v$ in $G$, using the final 'relationship-aware' embeddings, $\tilde{h}_{\tilde{T}}(\tilde{n}_{\mathbf{l}}^v)$, of $\tilde{n}_{\mathbf{l}}^v$ (purple nodes) in $\tilde{G}$. The variable nodes that are preassigned a value (non-zero yellow nodes) in $\mathbf{x}$, are initialized by the embedding of the corresponding value node, *i.e.*, if $\mathbf{x}[\mathbf{j}] = \mathbf{l}$, then $n_{\mathbf{j}}$ is initialized with the 'relationship-aware' embedding, $\tilde{h}_{\tilde{T}}(\tilde{n}_{\mathbf{l}}^v)$, of $\tilde{n}_{\mathbf{l}}^v$. The embedding of nodes corresponding to the unassigned variables ('-1' yellow nodes) are initialized by the average, $(1/k) \sum_{v_{\mathbf{l}} \in \mathcal{V}} \tilde{h}_{\tilde{T}}(\tilde{n}_{\mathbf{l}}^v)$, of all 'relationship-aware' embeddings. Initialize hidden state $h_0(n_{\mathbf{j}})$ of each variable node $n_{\mathbf{j}}$ with a $\mathbf{0}$ vector.

**Loss Computation:** For each variable represented by node $n_{\mathbf{j}}$, the ground truth value $\mathbf{y}[\mathbf{j}]$ acts as the target for computing standard Cross Entropy Loss. The probabilities over $\mathcal{V}$ are computed as follows: At step $t$, a scoring function, $s$, computes a score, $s(h_t(n_{\mathbf{j}}), h_t(n_{\mathbf{l}}^v))$, for assigning a value $v_{\mathbf{l}} \in \mathcal{V}$ to a variable $n_{\mathbf{j}}$ based on the hidden state of corresponding value and variable nodes. For each variable node, a $Softmax$ converts these scores into probabilities over the values $v_{\mathbf{l}} \in \mathcal{V}$, *i.e.*, $Pr(n_{\mathbf{j}}.v = v_{\mathbf{l}}) = Softmax(s(h_t(n_{\mathbf{j}}), h_t(n_{\mathbf{l}}^v))), \forall v_{\mathbf{l}} \in \mathcal{V}$, where, $n_{\mathbf{j}}.v \in \mathcal{V}$ denotes the value that node $n_{\mathbf{j}}$ can take. Loss at step $t$ is nothing but the average over variable nodes: $L_t = -\frac{1}{|N|} \sum_{n_{\mathbf{j}} \in N} \log Pr(n_{\mathbf{j}}.v = \mathbf{y}[\mathbf{j}])$. To ensure that the multi-valued model learns different embeddings for each value in the value-set, we add an auxiliary loss term, corresponding to the total pairwise dot product (similarity) of any two embeddings, before and after message passing in $\tilde{G}$. We call it *Orthogonality Loss*. Its weight, $\alpha$, is a hyper-parameter.

**Prediction on a problem with larger size of value-set:** For a puzzle with larger value-set, $\mathcal{V}'$, a bigger RMPG is created, whose $k'$ nodes are initialized with the (learnt) first $k'$ embeddings. Unlike training, we always choose first $k'$ embeddings to avoid randomness during testing. Prediction is made using the probabilities at the last step $T$, *i.e.*, $\hat{\mathbf{y}}[\mathbf{j}] = \arg\max_{v_{\mathbf{l}} \in \mathcal{V}'} Pr(n_{\mathbf{j}}.v = v_{\mathbf{l}})$.

**Relative Comparison:** In the binarized model, the constructed graph $G$ has $k|N_C|$ nodes and at least $k|E_C| + |N_C|k(k-1)/2$ edges due to binarization. This increases the graph size by a factor of at least $k$. As a result, we soon hit the memory limits of a GPU while training the binarized model with bigger problems. The model also needs significantly more inference time due to its bigger size. On the other hand, multi-valued model, while being compact in terms of its representation, needs to learn additional embeddings, for a speculative size of value-set during testing. This poses additional requirement on the model both in terms of representation, and learning, possibly affecting the quality of generalization. While this is a simple analytical understanding of the possible merits of the two models, we examine experimentally the impact of these issues on real datasets.

## 5 EXPERIMENTS

The goal of our experiments is to evaluate the effectiveness of our two proposed methods for achieving value-set invariance. We compare our models with a generic neural constraint learner, NLM (Dong et al., 2019). [2] We experiment on datasets generated from Lifted CSPs of three different puzzles, *viz.*, Sudoku, Futoshiki, and Graph Coloring (ref. Table 5 in appendix for details). We train each model on data generated from a fixed value-set, and test on instances generated from larger value-sets.

---

[2] Our aim is not to directly compete with SOTA SAT solvers, which are much more scalable than neural methods. Refer to appendix for a discussion on comparison with them as well as neural SAT solvers.

## 5.1 TASK DESCRIPTION AND DATASETS

**Futoshiki:** This is a number puzzle in which we have to place numbers $\{1 \ldots k\}$ on a $k \times k$ grid, such that no two cells in a row or column contain the same number. In addition, there may be an ordering constraint between two cells, which needs to be honored in the final solution. The input has some of the grid cells already filled with a number and the task is to complete the grid, respecting the additional ordering constraint where ever it exists. We train our model on $6 \times 6$ puzzles, with the percentage of missing cells varying uniformly between $28 - 70\%$. We test our models on puzzles with board size ranging between $6 \times 6$ to $12 \times 12$, with the same percentage of missing cells.

**Graph Coloring (GCP):** In this task we are given a partially colored graph along with the number of colors $k$, and the objective is to color rest of the nodes using $k$ colors such that no two adjacent nodes have the same color. We train our model on randomly generated $4-$colorable graphs, and test on $k'-$colorable graphs, with $k' \in \{4, 5, 6, 7\}$. Training data has graphs with graph order varying uniformly between $40 - 120$, and percentage of masked nodes vary uniformly between $28 - 70\%$.

Table 2: GCP: Mean (Std. dev.) of coloring and pointwise accuracy on graphs with different chromatic number.

| | Board Accuracy | | | |
|---|---|---|---|---|
| | **4** | **5** | **6** | **7** |
| **NLM24** | 81.34 (5.93) | 70.78 (7.45) | 71.25 (8.35) | 73.20 (7.58) |
| **MV** | 97.80 (0.03) | **97.72 (0.37)** | 94.03 (2.54) | 72.21 (11.17) |
| **BIN** | **99.09 (0.07)** | 96.69 (2.61) | **95.7 (4.04)** | **94.35 (4.82)** |
| | Pointwise Accuracy | | | |
| **NLM24** | 99.47 (0.13) | 98.58 (0.34) | 97.95 (0.54) | 97.26 (0.68) |
| **MV** | **99.96 (0.00)** | **99.89 (0.00)** | 99.50 (0.23) | 96.22 (1.55) |
| **BIN** | **99.96 (0.01)** | 99.85 (0.03) | **99.76 (0.08)** | **99.48 (0.16)** |

**Sudoku:** We randomly select $10,000$ training queries from the $9 \times 9$ dataset introduced in Palm et al. (2018). Our test set has $k' \times k'$ puzzles, with $k' \in \{10, 12, 15, 16\}$. Data generation process is similar to Futoshiki, with the distribution of missing cells varying between $30 - 68\%$ depending on the board size. Instead of backtracking, solution validity is checked through the GSS library (Pieters, 2019). Please see appendix for more details on data generation process for all three tasks.

## 5.2 EXPERIMENTAL SETUP & BASELINES

In both our models, nodes are initialized with learnable 96 dimensional embeddings. In multi-valued model, $k_{\max} = 12, 7$, and 16 embeddings are learnt for Futoshiki, GCP, and Sudoku respectively. Message passing on $G$ in binarized

Table 3: Sudoku: Mean (Std. dev.) of board and pointwise accuracy on different board-sizes. Both models trained on $9 \times 9$ puzzles

| | Board Accuracy | | | | |
|---|---|---|---|---|---|
| | **9** | **10** | **12** | **15** | **16** |
| **MV** | 92.78 (0.08) | 99.65 (0.15) | 88.30 (6.08) | 29.33 (13.71) | 19.70 (14.03) |
| **BIN** | **99.13 (0.14)** | **99.91 (0.04)** | **99.63 (0.10)** | **63.05 (45.71)** | **27.31 (23.81)** |
| | Pointwise Accuracy | | | | |
| **MV** | 98.52 (0.05) | 99.96 (0.02) | 99.43 (0.26) | **97.03 (0.71)** | **96.30 (0.90)** |
| **BIN** | **99.87 (0.02)** | **99.99 (0.00)** | **99.96 (0.01)** | 95.55 (6.60) | 88.39 (14.25) |

model runs for 32 steps. Message passing on RMPG, $\tilde{G}$ and CMPG, $G$ in the multi-valued model runs for $\tilde{T} = 1$ and $T = 32$ steps respectively. The message passing functions in both the models are 3 layer MLPs, similar to those in RRN, with a difference that there is a separate function for each edge type. In both the models, a layer normalized LSTM cell with hidden dimension 96 acts as state update functions. All models are trained on K40 GPU nodes with 12GB memory. We take simple average of model weights stored at multiple points (Izmailov et al., 2018). All checkpoints obtained after flattening of the learning curve are selected for computing average. See appendix for details.

**Baseline:** For Futoshiki, we train two versions of NLM by varying *depth*: the number of Logic Machines that are stacked on top of each other. Like (Nandwani et al., 2021), we train one 30 layer deep NLM model with residual connections for Futoshiki, but unlike them, we assume access to constraint graph, which we provide as a binary predicate input to the model. NLM with 30 depth could not fit puzzles with board-size greater than 10 within 12GB memory of K40 GPU. Hence, we train another version by reducing the depth to 15. For GCP, we train a model with depth 24. For Sudoku, on increasing depth beyond 14, we could not fit even one $9 \times 9$ train sample within GPU memory. Note that the maximum depth chosen for the graph experiments reported in (Dong et al., 2019) is 8. This is because they work with much smaller graphs (up to maximum 50 nodes), whereas smallest graph in Futoshiki has $6^3 = 216$ binary nodes, warranting creation of much deeper models.

**Evaluation Metrics:** We report two metrics: *board accuracy* and *point-wise accuracy*. In the former, we consider output of the model as correct only if it satisfies the underlying CSP, whereas in the later,

we give partial credit even for assigning some of the variables correctly. See Appendix for details. For each setting, we report the mean and standard deviation over three runs by varying random seed.

## 5.3 RESULTS AND DISCUSSION

We report the accuracies over different sizes of value-set for Futoshiki, GCP and Sudoku in Table 1, 2, and 3, respectively. We first observe that NLM fails to train on Sudoku, and its performance is worse than one or both of our models for all experimental settings in Futoshiki and GCP. As

Table 4: Sudoku: Mean (Std. dev.) of board and pointwise accuracy of models fine-tuned on 24 board-size

| | Board Accuracy | | | |
|---|---|---|---|---|
| | **15** | **16** | **24** | **25** |
| MV | **91.03 (3.25)** | **90.39 (3.49)** | **54.57 (21.25)** | **43.77 (14.42)** |
| BIN | 63.05 (45.71) | 27.31 (23.81) | 0.0 (0.0) | 0.0 (0.0) |
| | Pointwise Accuracy | | | |
| MV | **99.43 (0.16)** | **99.46 (0.15)** | **99.30 (0.12)** | **99.10 (0.09)** |
| BIN | 95.55 (6.60) | 88.39 (14.25) | 7.85 (0.63) | 7.44 (0.43) |

expected, in Futoshiki, NLM model with depth 30 fails to run on board sizes 11 and 12 and depth 15 model fails to run on size 12. Note that both NLM and our binarized model work by binarizing the underlying puzzle, but we observe that binarized model shows significantly better generalization across value-sets. We note that NLM performs decently well for GCP even for the test graphs with chromatic number $k' = 7$. We attribute this to the fact that in our test data for $k' = 7$, graphs are relatively small, with max 80 graph nodes, resulting in total 560 binary objects in NLM, which is similar to the max 400 binary objects that it trains over ($k=4$, max 100 nodes).

**Comparison between binarized model and multi-valued model:** We first observe that both our models achieve similar performance on the value-set over which they are trained. We observe that the standard deviation of the *board accuracy* increases significantly as the size of value-set increases, whereas the *pointwise accuracy* is relatively stable. This is due to the high sensitivity of the board accuracy to pointwise accuracy: even if a single variable is incorrectly assigned in a puzzle, its contribution towards board accuracy goes to 0, whereas it still contributes positively towards pointwise accuracy. When trained on small sizes, binarized model shows better generalization. But as the problem size increases, the computational graph for binarized model fails to fit in the available GPU memory and thus its performance degrades. On the other hand, multi-valued model being memory efficient, scales much better. To demonstrate this, Table 4 reports the performance of multi-valued model further finetuned on sudoku puzzles of board-size 24, and tested on board-sizes varying between 15 and 25. We couldn't finetune the binarized model as its computational graph doesn't fit in the GPU. The binarized model trained on puzzles of board-size 9 gives 0.0 board accuracy on size 24 and 25. The performance of multi-valued model is better than binarized model not only on board-size 25, but also on board-sizes smaller than 24. This also demonstrates that the poor performance of the same multi-valued model trained on smaller board-size is not due to any lack of *representation power*, but due to difficulty in learning additional embeddings: when training $k'$ embeddings from puzzles of board-size $k$, multi-valued model never gets to see all $k'$ value embeddings together. Moreover, the different combinations of $k$ out of $k'$ embeddings increase exponentially with $(k' - k)$, making it further difficult to train. To validate this, we train a multi-valued model with only 7 learnable embeddings for Futoshiki and observe that the board accuracy on 7 board-size increases to $97.82\%$ (at par with binarized model) from $90.18\%$ which is achieved when trained with 12 embeddings.

**Computational complexity:** In fig. 2, for the two models, we compare the average inference time and the GPU memory occupied by a batch of 32 Futoshiki puzzles over value-sets of varying sizes. As expected, the multi-valued model is much more efficient, both in terms of time and memory.

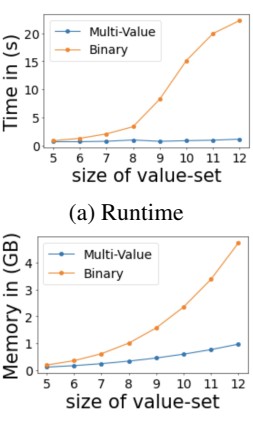

(a) Runtime

(b) Memory

Figure 2: Resource: Futoshiki

## 6 CONCLUSION AND FUTURE WORK

We have looked at the novel problem of value-set invariance in combinatorial puzzles, formally defined using lifted CSPs and proposed two different neural solutions extending RRNs. Our experiments demonstrate the superior performance of our models compared to an existing neural baseline. We discuss the relative strengths and weaknesses of our proposed models. Future work includes solving more complicated CSPs, and scaling to even larger sizes.

## ACKNOWLEDGEMENT

We thank IIT Delhi HPC facility[3] for computational resources. We thank anonymous reviewers for their insightful comments and suggestions that helped in further improving our paper. Mausam is supported by grants from Google, Bloomberg, 1MG and Jai Gupta chair fellowship by IIT Delhi. Parag Singla is supported by the DARPA Explainable Artificial Intelligence (XAI) Program with number N66001-17-2-4032. Both Mausam and Parag Singla are supported by the Visvesvaraya Young Faculty Fellowships by Govt. of India and IBM SUR awards. Any opinions, findings, conclusions or recommendations expressed in this paper are those of the authors and do not necessarily reflect the views or official policies, either expressed or implied, of the funding agencies.

## ETHICS STATEMENT

In its current form, our work is primarily a technical contribution, with no immediate ethical consequences. Our work develops the line of recent research in which constraint reasoning is carried out through neural architectures. We believe that neural approaches for symbolic reasoning will go a long way in creating an integrated AI system. This is because an integrated system requires not only perceptual, but also high-level reasoning. Neural approaches will provide a uniform vocabulary so that both these forms of reasoning can interact with each other, improving performance of the overall system.

As more AI systems start to be used in critical applications such as healthcare, law, and disaster management, it is important that they honor the safety and accountability constraints set up by domain experts. Their ability to perform high-level reasoning enables them to honor such constraints more effectively. Thus, our line of work, in the long run, could have significant positive ethical implications. We see no obvious negative implications of our work.

## REPRODUCIBILITY STATEMENT

To ensure reproducibility, we have discussed the dataset creation process and provided model architecture details in Section 5.1 and Section 5.2, respectively. We provide the details of the exact hyper-parameters, computational resources used, and additional experimental details in the appendix. We also make our code publicly available at *https://github.com/dair-iitd/output-space-invariance*.

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

## A  APPENDIX

## 2  RELATED WORKS

**Generalization of GNNs across graph size:** Our work relies heavily on the assumption that GNNs generalize across size. Here we briefly discuss the works that question the same. The existing set of papers (and results) in this line of research can be broadly divided into two sub-classes. The first set of results talk about the *representation power* of GNNs to handle various graph sizes. The second set of results talk about *learnability* issues with GNNs under varying train/test distributions. We look at some of the results below and try to explain why GNNs in our case are able to generalize well, both in terms of representation power, as well as learnability.

*Representation Power:* We hypothesize that there are two design choices that are helping us gain good representation power: 1. Ability to create a deep network without blowing up the number of parameters because of weight tying across layers, and 2. Preassigned class labels to some of the variables which act as node features and help in breaking the symmetry. We argue it on the basis of Theorem 4.2 in Yehudai et al. (2021), which proves that there exists a $(d + 3)$ layered GNN that can distinguish between nodes having different local structure, which is quantified via d-patterns that can be thought of as a generalization of node degree to d-hop neighborhood. Hence, to be able to distinguish between nodes on the basis of a GNN, all we need to do is ensure that different nodes have different d-patterns. This can be achieved in 2 ways: 1. By increasing $d$, e.g. two nodes may have the same degree and hence the same 1-pattern, but their neighbors may have different degrees, which will lead to different 2-pattern for these two nodes. 2. By assigning node features, e.g. two nodes may have the same degree, but their neighbors may have different node features, leading to a different 1-pattern for them as d-pattern also takes initial node features into account. In addition, Tang et al. (2020) also argue that one way of increasing the representation power of GNNs is by increasing their depth, and it achieves the same by proposing IterGNN that applies the same GNN layer for an adaptive number of iterations depending on the input graph. This is equivalent to tying the weights of different layers as in RRNs, as well as in our models.

*Learnability:* With respect to learnability, Yehudai et al. (2021) prove the existence of a 'bad' local minima that overfits on train data but fails on test samples that have unseen d-patterns. Our test dataset clearly has unseen d-patterns (e.g. nodes in 16 x 16 sudoku have different degrees than nodes in 9 x 9 sudoku), but our models still generalize. We note that Yehudai et al. (2021) only talks about the existence of some bad local minima, but does not rule out the possibility of the existence of other good local minima, which could generalize well, despite differences in local structure between train and test sets. This goes into the whole learnability argument, and whether we can find such not-so-bad local minimas (which presumably exist since the possibility has not been ruled out). One aspect that possibly comes to our rescue is that, unlike most GNN architectures, our design is recurrent in nature, i.e., parameters are tied across different GNN layers as inspired by Palm et al. (2018). Parameter tying assumption, possibly helps us in learnability, since the recurrence can be seen as a form of regularization, avoiding overfitting (or getting stuck in bad local minima). Exploring this further is a direction for future work.

In addition to Yehudai et al. (2021), Bevilacqua et al. (2021) deal with varying train/test distributions by proposing a size invariant representation of graphs. Their approach focuses on graph classification tasks, and is limited to creating size invariant representations for the entire graph. The theoretical claims presented in their paper primarily focus on the limitation of standard GNN based formulations for generalizing across sizes for graph classification tasks. On the other hand, we are interested in learning representations for each node in the graph for node classification, and it is not clear how the claims, as well as techniques proposed in the paper, extend to our setting.

## 4  MODELS DESCRIPTION

### 4.1  BINARIZED MODEL

**Recurrent Message Passing**

There are two categories of edges in the Message Passing Graph: *Constraint Edges* and *Relation Edges*. Each edge inherits an *edge type*, either from Constraint Graph, or Relation Graph. We denote

the set of all constraint edge types as $Q$, and the set of all relational edge types as $R$. We now describe the details of message passing and hidden state update equations.

**Edge Dependent Message Passing:** The nodes communicate their current hidden state via the messages sent to their neighbouring nodes across the edges. The message depends not only on the current state of the sender and receiver, but also on the *edge type* across which the message is sent. Specifically, for each edge type, $z$, there is a separate message passing function, $f_z$, with $z \in (Q \cup R)$ where $Q$ and $R$ are the set of all constraint edge types and relation edge types respectively. We compute the message for each edge $e^z_{(\mathbf{j_1 l_1}, \mathbf{j_2 l_2})} \in E$ as:

$$m_t \left[ e^z_{(\mathbf{j_1 l_1}, \mathbf{j_2 l_2})} \right] = f_z \left( h_t \left( n_{\mathbf{j_1, l_1}} \right), h_t \left( n_{\mathbf{j_2, l_2}} \right) \right), \forall e^z_{(\mathbf{j_1 l_1}, \mathbf{j_2 l_2})} \in E, \; z \in (Q \cup R).$$

**Hidden State Update:** For each node, the incoming messages on the edges of the same type are aggregated by taking their weighted average. The weights, $a_t$, are computed using Bahdanau Attention (Bahdanau et al., 2015) over constraint edges, whereas messages across relation edges are simply averaged: $m_{t,z}[n_{\mathbf{j,l_1}}] = \sum_{e^z_{(\mathbf{j l_1}, \mathbf{j_2 l_2})} \in E} a_t [e^z_{(\mathbf{j l_1}, \mathbf{j_2 l_2})}] m_t [e^z_{(\mathbf{j l_1}, \mathbf{j_2 l_2})}], \; \forall z \in (Q \cup R)$

Finally, all messages, $m_{t,z}[n_{\mathbf{j,l}}] \forall z \in (Q \cup R)$, are concatenated to create the input, $m_t[n_{\mathbf{j,l}}]$ for each node, $n_{\mathbf{j,l}}$. The hidden state at step $t$ is updated by the following state update function to generate the state $h_{t+1}(n_{\mathbf{j,l}})$: $h_{t+1}(n_{\mathbf{j,l}}) = g \left( h_t(n_{\mathbf{j,l}}), m_t[n_{\mathbf{j,l}}], u_0(n_{\mathbf{j,l}}) \right), \forall n_{\mathbf{j,l}} \in N$. See Figure 3 for an illustration of edge dependent message passing and state update at a given step $t$.

## 4.2 MULTI-VALUED MODEL

There are two separate message passing graphs in multi-valued model: RMPG and CMPG. RMPG contains edges encoding the relationship between the values. Each edge has an associated edge type representing the relationship it encodes. We denote the set of all edge types in RMPG as $R$. In CMPG, there are two categories of edges: *Constraint Edges* and *Assignment Edges*. Further, each edge may have an associated edge type. The set of all constraint edge types is denoted as $Q$, and the set of assignment edge types (edges from orange value nodes to yellow variable nodes in fig. 1) is denoted as $A$. Finally, the initial embedding of a variable node $n_{\mathbf{j}}$ is denoted as $u_0(n_{\mathbf{j}})$.

We now describe the message passing rules and hidden state update equations.

**Recurrent Message Passing ( on RMPG)**

**Message Passing Update:** At step $t$, update the hidden state, $\tilde{h}_t(\tilde{n}^v_{\mathbf{l}})$, of each of the value nodes in $\tilde{G}$, by the concatenation, $m_t[\tilde{n}^v_{\mathbf{l}}]$, of average messages, $m^r_t[\tilde{n}^v_{\mathbf{l}}]$, received across edges of type $r \in R$: $\tilde{h}_{t+1}(\tilde{n}^v_{\mathbf{l}}) = \tilde{g}(\tilde{h}_t(\tilde{n}^v_{\mathbf{l}}), m_t[\tilde{n}^v_{\mathbf{l}}], \tilde{u}[\mathcal{P}_{\mathbf{x}}[\mathbf{l}]])$, where $\tilde{g}$ is the hidden state update function. Like (Palm et al., 2018), it always takes the initial embedding, $\tilde{u}[\mathcal{P}_{\mathbf{x}}[\mathbf{l}]]$, of the value node $\tilde{n}^v_{\mathbf{l}}$ as one of the inputs. Notice that the message, $m^r_t[\tilde{n}^v_{\mathbf{l}}]$, is the average of the messages, $f_r(\tilde{h}_t(\tilde{n}^v_{\mathbf{l}}), \tilde{h}_t(\tilde{n}^v_{\mathbf{l'}})) \forall \tilde{e}^r_{(\mathbf{l}, \mathbf{l'})} \in \tilde{E}$, where $f_r$ is the message passing function for edge type $r \in R$. The hidden states are updated for $\tilde{T}$ steps and the final embeddings, $\tilde{h}_{\tilde{T}}(\tilde{n}^v_{\mathbf{l}}) \forall v_{\mathbf{l}} \in \mathcal{V}$, are used as 'relationship-aware' embeddings for initializing the input features (embeddings) of both variable nodes, $n_{\mathbf{j}}$, and value nodes, $n^v_{\mathbf{l}}$ in $G$ (orange and yellow nodes respectively in Multi-Valued Graph in fig. 1).

**Recurrent Message Passing ( on CMPG)**

**Message Passing Update:** At step $t$, similar to binarized model, each variable node receives messages from its neighbors, that are aggregated based on the edge type. For each node, the aggregated messages, $m_{t,z}[n_{\mathbf{j}}]$, for different edge types, $z \in (Q \cup A)$, are stacked to create, $m_t[n_{\mathbf{j}}]$, which updates the hidden state as: $h_{t+1}(n_{\mathbf{j}}) = g \left( h_t(n_{\mathbf{j}}), m_t[n_{\mathbf{j}}], u_0(n_{\mathbf{j}}) \right), \forall n_{\mathbf{j}} \in N$.

**Discussion on an alternate Encoding Scheme**

As discussed in section 4.1, the main intuition for our binarized modelcomes from 'sparse encoding' of an integer CSP to a SAT. In addition to 'sparse encoding', there is another way of converting integer CSP into SAT, called 'compact encoding' (Ernst et al., 1997; Iwama & Miyazaki, 1994), in which each Boolean SAT variable represents a single bit of the integer value that a CSP variable can take. The final assignment of a CSP variable is given by the integer represented by the $\log k$ Boolean

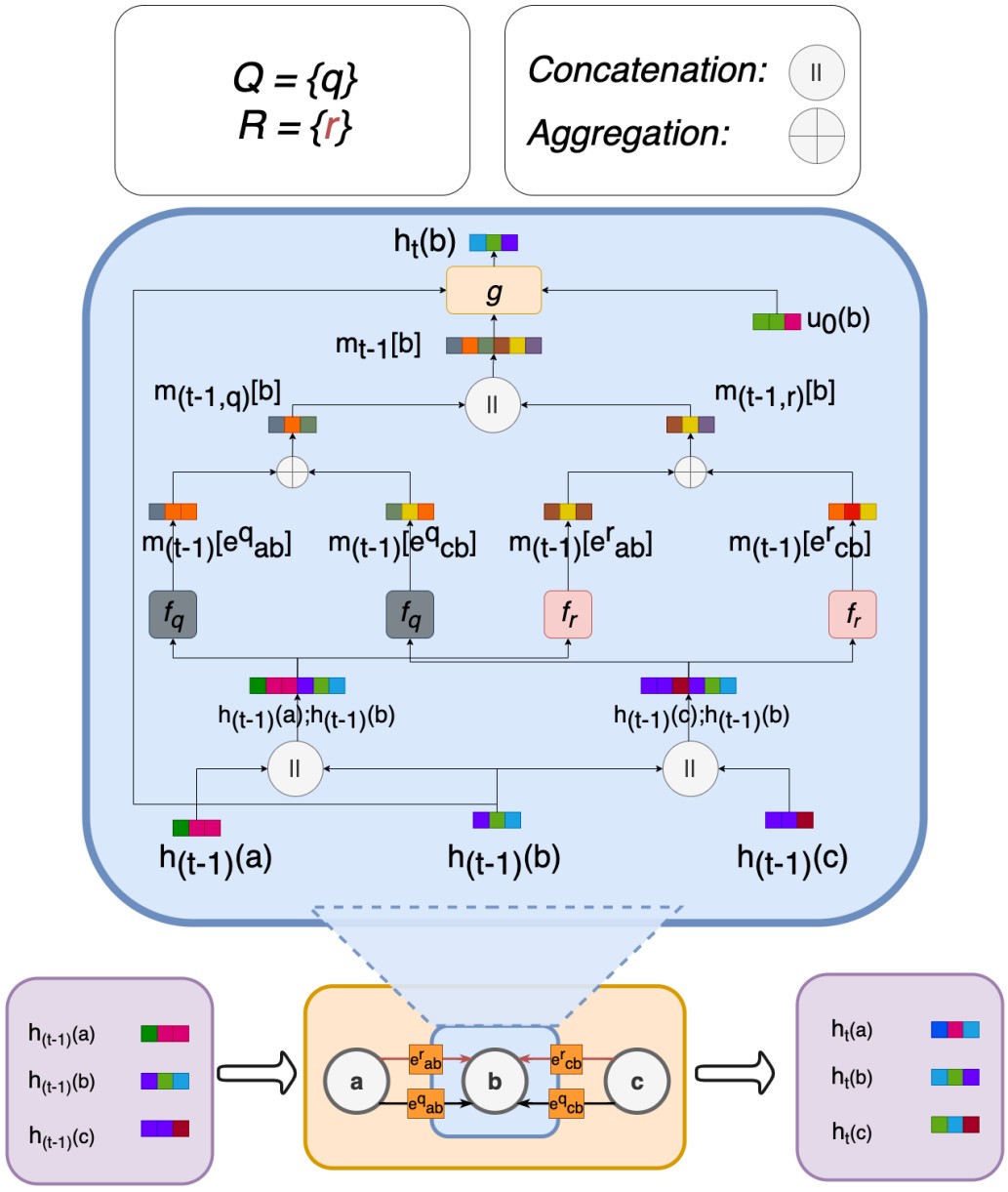

Figure 3: Hidden State Update at time-step $t$: We take a toy graph with 3 nodes, $\{a, b, c\}$, and 2 edge-types, $\{q, r\}$, to illustrate edge-dependent message passing and hidden state update of node $b$. First, messages along the four edges are calculated as an edge-type dependent function of the sender and receiver hidden state using $f_q$ and $f_r$. Next, the incoming messages are aggregated by edge-type (*e.g.*, using attention based mechanism or simple averaging), and the outputs are concatenated to obtain the final message, $m_{t-1}[b]$. The hidden state of node $b$ is updated by function $g$ which takes the previous hidden state $h^{t-1}(b)$, the incoming message $m_{t-1}[b]$, and the initial embedding of the node as its input.

SAT variables corresponding to that variable. Motivated by the 'compact encoding', one can construct another model: instead of a one-hot encoding which requires $k$ nodes (one for each value in $\mathcal{V}$) for each variable, create $\log k$ binary valued nodes for each variable and assign a value $v \in \mathcal{V}$ to the variable based on the integer represented by $\log k$ bits corresponding to it. This results in a graph with $|N_C| \log k$ nodes for a CSP with $|N_C|$ variables and $k$ classes, instead of $|N_C|k$ nodes in the binarized model, and brings it closer to the graph size of $|N_C| + k$ created in multi-valued model. However, such an approach failed to generalize across the size of the value-set in our experiments. In addition, such an encoding has a limitation in its representational capability. It can not encode the relationship between the values effectively. For example, in $k \times k$ Futoshiki, we have an ordinal relationship between the $k$ values representing the numerical numbers 1 to $k$. In our proposed approaches, we encode this by adding appropriate relational edges between nodes representing different values in $\mathcal{V}$. In the binarized model, it is done for each variable separately, whereas, in the multi-valued model, it is done in the RMPG. In absence of an explicit node for a value in this encoding scheme, it is not clear how to represent such a relationship.

# 5 EXPERIMENTS

**Discussion on comparison with SAT Solvers:** In this work, we are interested in creating (and learning) a neural solver for symbolic tasks, instead of using a symbolic algorithm like SAT solver. Such an approach has many benefits, *e.g.*, because of being differentiable, it can be used seamlessly in a unified framework requiring both perception as well as reasoning, e.g. visual sudoku; neural models have been shown to be resistant to varying amount of noise in the data as well, which purely logical (SAT style) solvers may not be able to handle. As is the case with other papers in this line of work e.g (Selsam et al., 2019; Amizadeh et al., 2019a), at this point our main motivation is scientific. We are interested in understanding to what extent neural reasoners can generalize across varying sizes of the value-set in train and test domains. Instead of comparing with an industrial SAT Solver, perhaps a fair comparison would be with a generic state-of-the-art neural SAT solver e.g., CircuitSAT (Amizadeh et al., 2019a), NeuroSAT (Selsam et al., 2019). Both of these papers observe that there is a long way to go before they can compete with industrial SAT solvers. In fact, both of these approaches experiment with much smaller problem instances. CircuitSAT uses a model trained on k-SAT-3-10 problems (k-SAT with 3 to 10 Boolean variables) for coloring graphs with number of nodes ranging between 6 to 10, and achieves a meager 27% performance and NeuroSAT fails to solve any of the problems in the coloring dataset used by CircuitSAT (section 5.2 in Amizadeh et al. (2019a)). On the other hand, the smallest of the problems in our dataset has 40 nodes (GCP) and the largest has $25^3 (= 25,625)$ nodes (in $25 \times 25$ Sudoku), and hence we do not expect the generic neural SAT solvers to scale up to our problem sizes.

Table 5: Dataset details

| Task | Train | | | | Test | | | |
|------|---|---------|----------|------------------|------|----------|----------|------------------|
| | k | #(Vars.) | Mask (%) | #(Missing Vars.) | k' | #(Vars.) | Mask (%) | #(Missing Vars.) |
| Futoshiki | 6 | 36 | 28-70 | 10-25 | {6,7,8,9,10,11,12} | 36-144 | 28-70 | 10-93 |
| GCP | 4 | 40-100 | 28-70 | 12-70 | {4,5,6,7} | 40-150 | 28-70 | 12-105 |
| Sudoku | 9 | 81 | 58-79 | 47-64 | {9,10,12,15,16} | 81-256 | 30-68 | 47-148 |
| Sudoku finetune | 24 | 576 | 30-70 | 173-403 | {15,16,24,25} | 225-625 | 30-70 | 68-314 |

## 5.1 TASK DESCRIPTION AND DATASETS

We experiment on datasets generated from Lifted CSPs of three different puzzles, *viz.*, Sudoku, Futoshiki, and Graph Coloring. In addition, we fine-tune our multi-valued model for Sudoku on 8000 puzzles of size 24 and test it on puzzles of different board sizes. Table 5 contains the details of both train and test data for the different experiments. Below we describe the three tasks and their datasets in detail.

**Futoshiki:** We train our model on $6 \times 6$ puzzles, with the percentage of missing cells varying uniformly between $28 - 70\%$. We test our models on puzzles with board size ranging between $6 \times 6$ to $12 \times 12$, with the same percentage of missing cells. The number of ordering constraints is twice the board size. To generate data, we first randomly generate a completely filled $k \times k$ board and then randomly mask $m\%$ of the cells. We search for its all possible completions using backtracking to

Table 6: Test Data statistics for all three tasks

| k | #Puzzles | #(Variables) | #(Missing Variables) | Mask (%) |
|---|---|---|---|---|
| **Futoshiki** | | | | |
| **6** | 4100 | 36 | 10-25 | 28-70 |
| **7** | 4091 | 49 | 14-34 | 29-70 |
| **8** | 3578 | 64 | 19-44 | 30-70 |
| **9** | 3044 | 81 | 24-56 | 30-70 |
| **10** | 2545 | 100 | 30-66 | 30-66 |
| **11** | 2203 | 121 | 36-82 | 30-68 |
| **12** | 1882 | 144 | 43-93 | 30-65 |
| **GCP** | | | | |
| **4** | 9102 | 40-150 | 12-105 | 28-70 |
| **5** | 9102 | 40-150 | 12-105 | 28-70 |
| **6** | 6642 | 40-120 | 12-84 | 28-70 |
| **7** | 3362 | 40-80 | 12-56 | 28-70 |
| **Sudoku** | | | | |
| **9** | 18000 | 81 | 47-64 | 58-79 |
| **10** | 2317 | 100 | 30-62 | 30-62 |
| **12** | 1983 | 144 | 43-84 | 30-58 |
| **15** | 1807 | 225 | 67-128 | 30-57 |
| **16** | 1748 | 256 | 76-148 | 30-58 |
| **24** | 1000 | 576 | 172-289 | 30-50 |
| **25** | 1000 | 625 | 187-314 | 30-50 |

ensure that it has only one solution. Finally, we insert ordering constraints between $2k$ randomly picked pairs of adjacent cells. The entire process is repeated by varying $k$ and $m$ to generate both the test and train dataset.

The training data consists of $12,300$ puzzles on 6 x 6 board size with the percentage of missing variables varying between $28 - 70\%$. The exact details of the testing data for different board sizes are provided in Table 6. We note that it becomes increasingly difficult to find puzzles with unique solution as the board size increases. Therefore, we are forced to reduce the maximum percentage of masked (unassigned) cells with increasing board size.

**GCP:** The training data for Graph Coloring Problem consists of around $25$ thousand 4-colorable graphs with graph order varying uniformly between $40 - 120$, and the percentage of unassigned (masked) variables varying uniformly between $28 - 70\%$ depending on the graph order. The exact details of the testing data for different chromatic number (value-set size) are provided in Table 6. To create non-trivial problems for the dataset, we always attempt to color graphs with the smallest possible number of colors, i.e., the chromatic number of the graph. We follow Erdős–Rényi (ER) model to generate random graphs. It takes number of nodes, $n$, and an edge probability, $p$, as input, and adds an edge independent of the other edges with probability $p$. We note that to sample a graph with $n$ nodes and a given chromatic number $k$, we need to carefully adjust the range from which edge probability $p$ is sampled. The exact range from which $p$ is sampled uniformly for each chromatic number $k$ and a range of nodes $n$ is given in Table 7. We use a CSP solver (Perron & Furnon, 2019) to determine the chromatic number of a given graph, which becomes a bottleneck while generating graphs with higher chromatic number. As a result, we were not able to generate graphs with more than $80$ nodes for chromatic number 7, in a reasonable amount of time.

**Sudoku:** The training data consists of $10$ thousand 9 x 9 puzzles randomly selected from the dataset introduced in (Palm et al., 2018). For standard $9 \times 9$ board, we use the same test data as used in RRN (Palm et al., 2018) [4]. The test data for the board-sizes between 10 and 16 is generated using the methodology similar to Futoshiki. Instead of backtracking, solution validity and uniqueness is checked through the GSS library (Pieters, 2019). The exact details of the testing data for different board sizes are provided in Table 6. For the experiment where we fine-tune our models on $24 \times 24$ puzzles, both the train and test data for board size 24 and 25 are generated following the methodology

---

[4]Available at: *https://data.dgl.ai/dataset/sudoku-hard.zip*

Table 7: Range for $p$ for given $k$ and $n$ for GCP data generation

| k \ n | 40-55 | 56-70 | 71-80 | 81-100 | 101-130 | 131-150 |
|---|---|---|---|---|---|---|
| 4 | (0.1, 0.2) | (0.05, 0.1) | (0.05, 0.1) | (0.05, 0.1) | (0.02, 0.05) | (0.02, 0.05) |
| 5 | (0.2, 0.25) | (0.1, 0.2) | (0.1, 0.2) | (0.075, 0.12) | (0.075, 0.1) | (0.05, 0.075) |
| 6 | (0.2, 0.25) | (0.15, 0.25) | (0.17, 0.2) | (0.15, 0.18) | (0.12, 0.16) | - |
| 7 | (0.325, 0.375) | (0.275, 0.325) | (0.22, 0.3) | - | - | - |

similar to Futoshiki. In this setup, we were not able to verify the uniqueness of solution through GSS library as it doesn't scale to such large sizes.

**Solution Multiplicity in GCP Dataset and larger board-size puzzles of Sudoku:** An input query in GCP may have more than one solution, out of which only one is given at train time. But the network may discover a new valid solution, and computing loss of the discovered solution w.r.t. the given solution may unnecessarily penalize the model. To avoid this, we algorithmically verify if a prediction is a valid coloring or not, and compute loss w.r.t. to this discovered solution, instead of the given one. This is equivalent to solving a One-of-Many Learning problem (Nandwani et al., 2021) with all possible colorings given at training time. The same phenomenon of solution multiplicity exists for Sudoku puzzles of size 24 and 25, as verifying the uniqueness of puzzles on such large board-size became computationally infeasible.

## 5.2 EXPERIMENTAL SETUP AND BASELINES

**Evaluation Metrics:** We report two metrics: *board accuracy* and *point-wise accuracy* for all our experiments. In the former, we consider output of the model as correct only if it satisfies the underlying CSP, whereas in the later, we give partial credit even for assigning some of the variables correctly. We formally define it below:

**Pointwise accuracy:** Let $Y_{\mathbf{x}}$ be the set of possible solutions for an input $\mathbf{x}$ with $k$ variables, and let $\hat{\mathbf{y}}$ be the model's prediction. Pointwise accuracy of the prediction $\hat{\mathbf{y}}$ with respect to the solution $\mathbf{y} \in Y_{\mathbf{x}}$, denoted as, $PointAcc(\mathbf{y}, \hat{\mathbf{y}})$, is defined to be the fraction of variables that match between the $\mathbf{y}$ and $\hat{\mathbf{y}}$: $PointAcc(\mathbf{y}, \hat{\mathbf{y}}) = \frac{1}{k} \sum_{i=1}^{k} \mathbb{1}\{\mathbf{y}[i] == \hat{\mathbf{y}}[i]\}$, where $\mathbb{1}\{.\}$ is the indicator function.

Given above, we define pointwise accuracy for a prediction $\hat{\mathbf{y}}$ of an input $\mathbf{x}$ with respect to a solution set $Y_{\mathbf{x}}$ to be the maximum among the pointwise accuracy with respect to each of the solutions in the set $Y_{\mathbf{x}}$. Mathematically, $PointAcc(Y_{\mathbf{x}}, \hat{\mathbf{y}}) = max_{\mathbf{y} \in Y_{\mathbf{x}}} PointAcc(\mathbf{y}, \hat{\mathbf{y}})$.

For Sudoku and Futoshiki, since there is a unique solution, we can easily compute pointwise accuracy as the target set $Y_{\mathbf{x}}$ is singleton. For the GCP task, whenever the model returns a valid coloring, pointwise accuracy is 1, otherwise, in the absence of access to complete $Y_{\mathbf{x}}$, we report a lower bound by performing a local search, using Google's OR-Tools [5], for a valid coloring closest to the model prediction. Same is true for sudoku puzzles on 24 and 25 board size.

**Why care about point-wise accuracy?** In our settings, the generalization problem can be hard, especially when there is a large difference between the sizes of the value-sets for the train and test domains. Given that we are defining a novel task, and it is important to measure progress even when problems are hard, we compare the two models using a simpler metric (pointwise accuracy) as well, in addition to board accuracy. This additional metric can help us detect progress, and also compare the relative performance of underlying models.

**Computational resources:** All models are trained on K40 GPUs with 12 GB memory, available on an HPC cluster.

**Hyperparameters:** The list below enumerates the various hyperparameters with a brief description and the values chosen.

1. **Batch Size:** For each task, we selected the maximum batch size that can be accommodated in 12GB GPU memory. Refer to Table 8 for details.

---

[5]https://developers.google.com/optimization

Table 8: Hyperparameters for different models and tasks

| Model | Batch Size | Weight Decay | Orthogonality Loss Factor | Edge Dropout |
|---|---|---|---|---|
| Futoshiki | | | | |
| MV | 64 | 0.0001 | 0.01 | 0.1 |
| BIN | 16 | 0.0001 | - | 0.1 |
| NLM15 | 4 | 0.0001 | - | - |
| NLM30 | 2 | 0.0001 | - | - |
| GCP | | | | |
| MV | 64 | 0.0001 | 0.01 | 0.1 |
| BIN | 16 | 0.0001 | - | 0.1 |
| NLM24 | 1 | 0.0001 | - | - |
| Sudoku | | | | |
| MV | 28 | 0.0001 | 0.01 | 0.1 |
| BIN | 3 | 0.0001 | - | 0.1 |

Table 9: Training cost of different models in terms of number of epochs, gradient updates and clock time

| Model | Batch Size | Training Data Size | # Gradient Updates | #Epochs | Time per Epoch (min) | Total Time (Hours) |
|---|---|---|---|---|---|---|
| Futoshiki | | | | | | |
| MV | 64 | 12,300 | 60,000 | 312 | 5 | 25 |
| BIN | 16 | 12,300 | 37,500 | 49 | 26 | 21 |
| NLM15 | 4 | 12,300 | 155,000 | 50 | 34 | 43 |
| NLM30 | 2 | 12,300 | 232,500 | 38 | 87 | 66 |
| GCP | | | | | | |
| MV | 64 | 25,010 | 80,000 | 205 | 10 | 33 |
| BIN | 16 | 25,010 | 40,000 | 26 | 39 | 17 |
| NLM24 | 1 | 25,010 | 260,000 | 10 | 213 | 37 |
| Sudoku | | | | | | |
| MV | 28 | 10,000 | 162,000 | 454 | 9 | 68 |
| BIN | 3 | 10,000 | 168,000 | 50 | 74 | 63 |

2. **Optimizer:** To minimize the loss, we use Adam optimizer with learning rate 0.0002. As in the original RRN paper, we chose a weight decay factor of 1E-4.

3. **Orthogonality Loss Factor:** To ensure that the multi-valued model learns different embeddings for each value in the value-set, we add an auxiliary loss term, corresponding to the total pairwise dot product of any two embeddings, before and after message passing on the Relation Message Passing Graph (RMPG), $\tilde{G}$. Its weight, $\alpha$, was chosen amongst $\{0.01, 0.1, 0.5\}$ by cross validating on a devset for Futoshiki, and then fixed afterwards for all our experiments.

4. **Edge Dropout:** While collating the messages from the edges of the same type, we drop 10% of the messages, as done in RRN. Dropout is used in the Message Passing Graph (MPG) of the binarized model, and the Constraint Message Passing Graph (CMPG) of the multi-valued model.

**Model Averaging:** As suggested in (Izmailov et al., 2018), to reduce the variance of our model performance, we take simple average of model weights stored at multiple points. All checkpoints beyond a point when the learning curve flattened are selected for computing the average.

**Training Time:** Table 9 enumerates the exact training cost, in terms of total training epochs, number of gradient updates, and clock time, for all three tasks and for both our models as well as the baseline

NLM model. Note that while a multi-valued model may have fewer parameters, and results in a much smaller graph and inference time for a given problem, its training time could still be higher, especially in terms of total training epochs and number of gradient updates. However, because of its memory efficiency, we can keep a much larger batch size during training, and because of its speed efficiency, each update is much faster. As a result, the overall clock time comes out to be comparable to the binary model for the two of our tasks, i.e. Futoshiki and Sudoku, and it is within 2x for GCP, even though the number of epochs is much higher.

