# OpenReview forum: "Neural Models for Output-Space Invariance in Combinatorial Problems"
_ICLR.cc/2022/Conference — ICLR 2022 Poster_

### Official Review · Reviewer_FY1P · 2021-11-01

**Correctness:** 3
**Technical Novelty And Significance:** 3
**Empirical Novelty And Significance:** 3
**Recommendation:** 8
**Confidence:** 3

**Main Review:**

Pros:

1. The problem itself is very essential since fixed output space does harm the generalization ability of the current neural solver. So a method that potentially can be applied to most neural solvers (based on message passing) is important.
2. The performance looks great to me. The multi-valued method largely outperforms the baseline and can generalize to big problem instances (like 24 by 24 Sudoku).
3. The training method of the multi-valued method is elegant.

Cons:

1. It would be better if the authors can try their method on the routing problems or other graph problems. Sudoku and Futoshiki are very similar. Moreover, graph coloring shares a similar property to Sudoku (basically all heavily rely on the All-Diff constraints).
2. I am wondering what's the training time/cost compared between the two methods. The binarized model should have more parameters but it should learn faster while the multi-valued model is hard to train especially when the target output space is very large. When the size of the problem goes very high, I feel the converge of the multi-valued model could be a problem.

Minors:

1. For generating the Sudoku larger than 9. It would be better if you make sure all the data are permutation invariant.

**Summary Of The Paper:**

This paper intends to develop a learning framework (based on the recurrent relational neural network (RRN) ) that can handle output-space invariance issue.  Output-space invariance means that the model trained on smaller size data (like 9 by 9 Sudoku) can be generalized to larger size data (16 by 16) without any further modification. The entire problem formulation is done by lifted CSP.

Based on RRN, the authors propose two methods: The binarized method and the Multi-value method.

For the binarized method: the output value space is binarized, i.e., the model will score each candidate individually and the parameters are shared.

For the multi-value model, the value nodes are added to the architecture and the nodes connected to value k mean that the node can be assigned value k or it is already assigned value k.  A decent method is proposed to help the model learn a meaningful representation for all the possible values.



**Summary Of The Review:**

This paper proposes two methods for the output-invariance issue based on the RRN framework. An elegant training method for one of the methods (the multi-valued method) is proposed. The authors tested their methods on three problems and achieve very good performance. Though the three test problems are similar, the problem itself is very novel and essential. So I would recommend for accpet.

---

> ### Author Response · Authors · 2021-11-13
> **Official response to Reviewer FY1P**
>
> Dear Reviewer FY1P,
>
> We thank you for an encouraging review of our manuscript. Below we answer your questions:
>
> **Training time comparison:** We agree with your observation. While a multi-valued model may have fewer parameters, and results in a much smaller graph and inference time for a given problem, its training time could still be higher, especially in terms of total training epochs and number of gradient updates. We have also briefly discussed this learning difficulty when we compare our two models in Section 5 (page 9, just above fig 2 in the original submission). However, because of its memory efficiency, we can keep a much larger batch size during training ( table 8 in appendix), and because of its speed efficiency, each update is much faster. As a result, the overall clock time comes out to be comparable to the binary model for the two of our tasks, i.e. Futoshiki and Sudoku, and it is within 2x for GCP,  even though the number of epochs is much higher. In the updated draft, we have added a table (table 9) in the appendix, comparing the exact training cost, in terms of total training epochs, number of gradient updates, and clock time.
>
> **Permutation invariance of the data in Sudoku:**
> We are not sure if we completely followed this comment.
> Do you mean that for any permutation of the digits, the training dataset should remain the same? That is, for a given training query, all its permutations should be in the training data. We would like to note that our binary model is permutation equivariant: a query and all its permutations would result in isomorphic graphs, resulting in the predictions that are permutations of each other. On the other hand, the training procedure of our multi-valued model first selects a random $k-$permutation of $k_{max}$ embeddings, which is equivalent to transforming the query with the same $k-$permutation.  Hence, the training of both models would not get affected by an arbitrary permutation of the training dataset. If we did not understand your question/comment correctly, we request you to revert back with additional details, and we will be happy to respond.
>
> **Experiments on other graph problems:**
> Although there are other constraints like ‘less than’ in Futoshiki, we agree that our experimental domains rely heavily on the all-diff constraints. In the future, we will experiment with more types of constraints. If you have suggestions for specific tasks, please share. Ideal testbeds are those tasks for which neural models have shown to be effective for problems with the same value-set size (e.g. RRN already trains and tests on 9 x 9 sudoku) so that we can focus on value-set generalization (i.e., testing on 16 x 16 sudoku). We also note that unless train/test datasets are readily available, it will require us to curate data, which can be time-consuming for typical NP-hard problems. With respect to the routing problems, we would like to note that the neural solutions to them are typically autoregressive, trained via RL instead of direct supervision over nodes in the graph (Kool et al. [ICLR 2019]), and it is not clear how value-set invariance comes into play in such problems.
>
> **References**
>
> [Kool et al. 2019] Attention, Learn To Solve Routing Problems!, ICLR 2019

---

### Official Review · Reviewer_XpUt · 2021-11-03

**Correctness:** 4
**Technical Novelty And Significance:** 3
**Empirical Novelty And Significance:** 3
**Recommendation:** 6
**Confidence:** 3

**Main Review:**


**Strengths:**
- An interesting problem that was not studied before
- The suggested ways of encoding the values in the graph make sense and are elegant.
- Empirical results seem good

**Weaknesses**

- The “size generalization works” assumption - the method assumes that GNNs can naturally generalize to larger graph sized. This is not necessarily true as was pointed out by several recent papers [“Size-Invariant Graph Representations for Graph Classification Extrapolations”, “From local structures to size generalization in graph neural networks”]. It would strengthen the paper if the authors could explain why this assumption works in their case.
- Choice of graph connectivity - I am missing a discussion on the different choices of encoding CSPs as graphs. It seems like the authors come up with these two formulations and I am wondering whether there are other choices of encoding this which are better. A discussion on this design space can help.
- Writing style - the main technical part of the paper, in which the graph constructions and the message passing are described, is written in a very dense way and is difficult to read. I think that a revision to these sections can improve the paper. Moreover, more illustrations can definitely help (for example illustrating the message passing schemes). The current figure is nice but not enough.




**Summary Of The Paper:**

The paper raises an overlooked problem when using GNNs to solve combinatorial puzzles such as Soduku and graph coloring. The problem is as follows: when given larger instances of the problem, the GNN will have to predict values from a larger value set. The question is how can we train on small instances and generalize to large instances with different output labels. It is important to note that the paper does not study the standard size-generalization problem (can GNNs generalize to larger graphs) but rather to a different related problem when the larger problem has more output labels.
The authors first formulate combinatorial puzzles as constraint satisfaction problems (CSP) and then suggest two ways of encoding these CSPs as graphs and process these graphs with GNNs. The idea is to encode the variable set of values in the graph structure. The first encoding is binarization - duplicating node variables for each possible output value. In the suggested encoding, called multiple values, they construct a graph with special nodes that represent the different possible values. Prediction is done by considering both variable and value node features together.
In the final section, the authors present results on 4 different combinatorial puzzles, where their method is shown to outperform a generic neural reasoner.


**Summary Of The Review:**

The paper targets a new interesting problem and suggests two elegant ways to solve it. There are, however, several aspects in which the paper can improve, including explaining the underlying assumptions and the design choices better.

**More comments**
- Intro: define CSP CNF  constraint graph
- figure 1 - add more details about the problems, constraints, and representation. Currently, it is difficult to understand.
- Lifted CSP formulation is difficult to understand. What do you mean by reference? After reading it several times I think I understand what you mean but I think this part can benefit from a revision. Perhaps add an illustration?
- Description of networks and message passing: too dense. Difficult to read. Visualization must be added.


**Post rebuttal:**
I would like to thank the authors for addressing my concerns. The paper has definitely improved, but I would like to maintain my score of 6 for now.

---

> ### Author Response · Authors · 2021-11-13
> **Official response to Reviewer XpUt (Part 1)**
>
> Dear Reviewer XpUt,
>
> We thank you for your valuable suggestions towards improving our paper by adding more discussion on the generalization assumptions and design choices. As suggested by you, we are also working on adding more illustrations and clarity in the technical section and will soon upload a new draft. We will also add a discussion section about the design choices and assumptions, either in the main paper or in the appendix depending on the space constraints. We briefly discuss both of them below:
>
> **Design Choices:**
>
> We would like to refer the reviewer to the “**Relative Comparison**” paragraph of section 4.2 (on page 7) where we theoretically discuss the pros and cons of the two proposed approaches. For the design choices in the multi-valued model, we refer the reviewer to “**Achieving value-set Invariance**” paragraph in section 4.2 (on page 6) which briefly discusses why we need to construct two different message-passing graphs (CMPG and RMPG).
>
> In addition to our two proposed approaches, we also experimented with another way of constructing the graph for encoding the CSPs. We call it ‘log model’.  Instead of a one-hot encoding which requires $k$ nodes (one for each class) for each variable as done in our binary-model, we created $log \ k$ binary valued nodes for each variable and classified the variable into $k$ classes based on the integer represented by total $log \ k$ bits. This results in a graph with $n log \ k$ nodes for a CSP with $n$ variables and $k$ classes, instead of $nk$ nodes in the binary model, and brings it closer to the graph size of $n+k$ created in multi-valued model. However, such an approach failed to generalize across the size of the value-set in our experiments. In addition, we also identified a limitation in the representational capability of log model. It can not encode the relationship between the classes effectively. For example, in $k \times k$ Futoshiki, we have an ordinal relationship between the $k$ classes representing the numbers $1$ to $k$. In our current approaches, we encode this by adding appropriate edges between nodes representing different classes. In the binary model, it is done for each variable separately, whereas, in the multi-valued model, it is done in the RMPG. In absence of an explicit node for a class in the log model, it is not clear how to represent such a relationship.
>
> **Why does GNN generalize for our tasks?**
>
> We thank you for pointing out relevant references that question and analyze the generalization capabilities of GNNs. The existing set of papers (and results) in this line of research can be broadly divided into two sub-classes. 1. The first set of results talk about the representation power of GNNs to handle various graph sizes. 2. Second set of results talk about learnability issues with GNNs under varying train/test distributions. We look at some of the results below and try to explain why GNNs in our case are able to generalize well, both in terms of representation power, as well as learnability.
>
> *Representation Power:* We hypothesize that there are two design choices that are helping us gain good representation power: 1. Ability to create a deep network without blowing up the number of parameters because of weight tying across layers, and 2. Preassigned class labels to some of the variables which act as node features and help in breaking the symmetry. We argue it on the basis of Theorem 4.2 in Yehudai et al. [ICML 2021], which proves that there exists a (d+3) layered GNN that can distinguish between nodes having different local structure (local structure is quantified via  d-patterns that can be thought of as a generalization of node degree to d-hop neighborhood). Hence, to be able to distinguish between nodes on the basis of a GNN, all we need to do is ensure that different nodes have different d-patterns. This can be achieved in 2 ways: 1. By increasing d, e.g. two nodes may have the same degree and hence the same  1-pattern, but their neighbors may have different degrees, which will lead to different 2-pattern for these two nodes. 2. By assigning node features, e.g. two nodes may have the same degree, but their neighbors may have different node features, leading to a different 1-pattern for them as d-pattern also takes initial node features into account.
> In addition, Tang et al. [Neurips 2020] also argue that one way of increasing the representation power of GNNs is by increasing their depth, and it achieves the same by proposing IterGNN that applies the same GNN layer for an adaptive number of iterations depending on the input graph. This is equivalent to tying the weights of different layers as in RRNs, as well as in our models.
>
> *continued in the next comment ...*

---

> > ### Author Response · Authors · 2021-11-13
> > **Official response to Reviewer XpUt (Part 2)**
> >
> > *continued from the previous comment ...*
> >
> > *Learnability:* With respect to learnability,  Yahudai et al. prove the existence of a ‘bad’ local minima that overfits on train data but fails on test samples that have unseen d-patterns. Our test dataset clearly has unseen d-patterns (e.g. nodes in 16 x 16 sudoku have different degrees than nodes in 9 x 9 sudoku), but our models still generalize. We note that Yehudai et al. only talks about the existence of some bad local minima, but does not rule out the possibility of the existence of other good local minima, which could generalize well, despite differences in local structure between train and test sets. This goes into the whole learnability argument, and whether we can find such not-so-bad local minimas (which presumably exist since the possibility has not been ruled out). One aspect that possibly comes to our rescue is that, unlike most GNN architectures, our design is recurrent in nature, i.e., parameters are tied across different GNN layers as inspired by Palm et al. [Neurips 2018]. Parameter tying assumption, possibly helps us in learnability, since the recurrence can be seen as a form of regularization, avoiding overfitting (or getting stuck in bad local minima). Exploring this further is a direction for future work.
> >
> > In addition to Yehudai et al., Bevilacqua et al. [ICML 2021] deal with varying train/test distributions by proposing a size invariant representation of graphs. Their approach focuses on graph classification tasks, and is limited to creating size invariant representations for the entire graph. The theoretical claims presented in their paper primarily focus on the limitation of standard GNN based formulations for generalizing across sizes for graph classification tasks. On the other hand, we are interested in learning representations for each node in the graph for node classification, and it is not clear how the claims, as well as techniques proposed in the paper, extend to our setting.
> >
> > In the end, we thank you again for your valuable suggestions. We will soon incorporate them in the manuscript.
> >
> >
> > **References**
> >
> > [Yehudai et al. 2021] From Local Structures to Size Generalization in Graph Neural Networks, ICML 2021
> >
> > [Tang et al. 2020] Towards Scale-Invariant Graph-related Problem Solving by Iterative Homogeneous Graph Neural Networks, NeurIPS 2021
> >
> > [Bevilacqua et al. 2021] Size-Invariant Graph Representations for Graph Classification Extrapolations, ICML 2021
> >
> > [Palm et al. 2018] Recurrent Relational Networks, NeurIPS 2018

---

### Official Review · Reviewer_VvZR · 2021-11-08

**Correctness:** 2
**Technical Novelty And Significance:** 3
**Empirical Novelty And Significance:** 2
**Recommendation:** 5
**Confidence:** 3

**Main Review:**

The problem of generalizing across the size of the output space (the value-set) seems important. However, I'm not sure that the authors succeed in their stated goal of "implicitly learning underlying constraints using only solved instances". In fact the Constraint Graph $G_C$ and the Relation Graph $G_R$ (defined in section 3), derived from lifted CSP $\mathcal{C}_L$ seems to be going a long way towards explicitly describing the underlying constraints. At this point, the amount of effort in describing the graph seems comparable to the amount of effort that would have to be put in to actually creating a SAT instance of the problem (which could be solved by a SAT solver).
Other than this, the paper is reasonable well written, and the approach that they are taking seems reasonable if the goal is to understand how much we can push GNNs to solve combinatorial problems (even if it does not meet the stated goal of not explicitly specying the constraints).
On the experimentation front, it feels like the the only real metric for a combinatorial problem is "board accuracy", and I did not completely understand the "pointwise accuracy" metric - how does one decide if a variable is correct if the overall solution is incorrect for say graph coloring? How is this defined in situations where the values could be permuted (like the colors) or when there are multiple correct solutions?



**Summary Of The Paper:**

The authors propose a (actually two) solution for solving combinatorial puzzles (such as Sudoku or graph coloring) by learning implicit constraints, using only solved instances in a way that generalizes across the size of the output space from which the variables are assigned. They exploit the graph size invariance of GNNs and convert a multi class node classification problem into a binary node classification problem, and use a generalization of the message passing approach of Palm et al. They show experimental results for their two solutions, and discuss the reasons for the accuracy / scalability tradeoff across the two.


**Summary Of The Review:**

The paper is well written, the approach makes sense, but it does not feel like they are actually accomplishing their goal of not explicitly encoding the constraints (and learning purely from the solved instances). Given how much they are actually encoding the constraints into the graph, it feels like they should at least compare it with other solutions which would require an equal amount of constraint encoding (such as using a SAT solver).
Further, the results of the algorithm are not particularly impressive, if we consider "board accuracy" (pointwise accuracy feels like the wrong metric for combinatorial problems).
On the pro side, this paper does address a problem that I've not seen tackled before and that seems to be important.

---

> ### Author Response · Authors · 2021-11-13
> **Official response to Reviewer VvZR (Part 1)**
>
> Dear Reviewer VvZR,
>
> We thank you for raising some important concerns. Below we discuss them:
>
> **Specifying constraint graph vs explicitly describing the underlying constraints:**
> We agree that providing the constraint graph does require additional manual efforts, but it is still not as much as specifying the constraints themselves. For instance, in Futoshiki, we have different kinds of constraints, such as diff, less-than, more-than, etc. and the constraint graph only needs to know which two nodes (cells) have a constraint between them. Further, it has been argued by the previous work (Palm et al. [NeurIPS 2018]), that similar results could be obtained by connecting all pairs of nodes, for example, in the case of Sudoku, though no explicit experimental evaluation has been done as such.
>
> **Comparison with SAT Solvers:**
> In this work, we are interested in creating (and learning) a neural solver for symbolic tasks, instead of using a symbolic algorithm like SAT solver. Such an approach has many benefits, e.g. because of being differentiable, it can be used seamlessly in a unified framework requiring both perception as well as reasoning, e.g. visual sudoku; neural models have been shown to be resistant to varying amount of noise in the data as well, which purely logical (SAT style) solvers may not be able to handle. As is the case with other papers in this line of work (e.g Selsam et al. [ICLR 2019], Amizadeh et al. [ICLR 2019]), at this point our main motivation is scientific. We are interested in understanding to what extent neural reasoners can generalize across varying sizes of the value-set in train and test domains. Instead of comparing with an industrial SAT Solver, perhaps a fair comparison would be with a generic state-of-the-art neural SAT solver e.g., CircuitSAT (Amizadeh et al. [ICLR 2019]), NeuroSAT (Selsam et al. [ICLR 2019]). Both of these papers observe that there is a long way to go before they can compete with industrial SAT solvers. In fact, both of these approaches experiment with much smaller problem instances. CircuitSAT uses a model trained on k-SAT-3-10 problems (k-SAT with 3 to 10 boolean variables) for coloring graphs with number of nodes ranging between 6 to 10, and achieves a meager 27% performance (section 5.2 in Amizadeh et al.); NeuroSAT fails to solve any of the problems in the coloring dataset used by CircuitSAT (section 5.2 in Amizadeh et al.). On the other hand, the smallest of the problems in our dataset has 40 nodes (GCP) and the largest has $25^3 (=25,625)$ nodes (in 25 x 25 sudoku), and hence we do not expect the generic neural SAT solvers to scale up to our problem sizes.
>
> **Pointwise accuracy:**
> Thanks for your comment regarding the need for a more precise definition of pointwise accuracy, especially when multiple solutions are present. We missed this in the original draft due to an overlook, and we have added it in the updated draft. Below, we first define pointwise accuracy, and then we describe what we have reported in the paper, followed by a justification for why this is a relevant metric to look at in addition to board-accuracy.
>
> *Pointwise accuracy:* Let $Y_x$ be the set of possible solutions for query $x$ with $k$ variables, and let $\hat{y}$ be the model’s prediction. Pointwise accuracy of the prediction $\hat{y}$ with respect to the solution $y \in Y_x$, denoted as, $PointAcc(y,\hat{y})$, is defined to be the fraction of variables that match between the $y$ and $\hat{y}$:
> $PointAcc(y,\hat{y}) = \frac{1}{k}\sum_{i=1}^{k} \mathbb{1}(y[i] == \hat{y}[i])$, where $\mathbb{1}(.)$ is the indicator function.
>
> Given above, we define pointwise accuracy for a prediction $\hat{y}$ for a query $x$ with respect to a solution set $Y_x$ to be the maximum among the pointwise accuracies with respect to each of the solutions in the set $Y_x$. Mathematically,
> $PointAcc(Y_x,\hat{y}) = max_{y \in Y_x} PointAcc(y,\hat{y}) $.
>
> For Sudoku and Futoshiki, since there is a unique solution, we can easily compute pointwise accuracy as the target set $Y_x$ is singleton.
> For the GCP task, whenever the model returns a valid coloring, pointwise accuracy is 1, otherwise, in the absence of access to complete $Y_x$, we have reported a lower bound by computing it with respect to the first coloring returned by a symbolic solver. The same is true for Sudoku puzzles on 24 and 25 board sizes.
>
> After your comment, in the updated draft, we have reported a tighter bound by doing a local search for a valid coloring closest to the model prediction.  As expected, this results in improvement (mostly marginal) of the pointwise accuracy numbers for GCP (Table 2) and Sudoku models finetuned on 24 board size puzzles (Table 4) but does not change the relative ordering of the different methods.
>
> *Continued in the next comment....*

---

> > ### Author Response · Authors · 2021-11-13
> > **Official response to Reviewer VvZR (Part 2)**
> >
> > *continued from the previous comment ...*
> >
> > **Why care about point-wise accuracy?** In our settings, the generalization problem can be hard, especially when there is a large difference between the sizes of the value-sets for the train and test domains. Given that we are defining a novel task, and it is important to measure progress even when problems are hard, we compared the two models using a simpler metric (pointwise accuracy) as well, in addition to board accuracy. This additional metric can help us detect progress, and also compare the relative performance of underlying models. It is good to note that in most of our experiments, board accuracy is monotonic with pointwise accuracy.
> >
> > **Quality of the results:**
> > We agree that actual accuracies obtained by our models are somewhat on the lower side and there is much scope for improvement. This is especially true when we try to generalize to large board sizes. At the same time, we would like to stress that (as already pointed out by you), we have defined a novel problem of value-set invariance in neural combinatorial solvers and proposed two solutions for the same. Our work can be seen as a first step in this direction, and we expect that our paper will generate/invigorate interest in the community to work on this line of research, resulting in improved solutions in the future.
> >
> > Finally, we thank you again for your valuable comments. We have uploaded a new draft with clarification on pointwise accuracy in the appendix.  We will soon update the manuscript with a short discussion on comparison with SAT Solvers.
> >
> > **References**
> >
> > [Amizadeh et al. 2019] Learning to solve circuit-sat: An unsupervised differentiable approach, ICLR 2019
> >
> > [Selsam et al. 2019]  Learning a SAT solver from single-bit supervision, ICLR 2019
> >
> > [Palm et al. 2018] Recurrent Relational Networks, Neurips 2018

---

### Author Response · Authors · 2021-11-21
**Summary of changes in the manuscript (Common response to all the reviewers)**

Dear Reviewers,

We thank you once again for your insightful comments that have helped us improve our paper. We have uploaded a revised version of our manuscript after incorporating your valuable suggestions, either in the main paper or in the appendix, depending on the space constraints.
Please let us know if any of the comments are still unaddressed, and we will be happy to incorporate them.

Specifically:

1.  We have revised section 4 significantly to add more clarity. In addition, we have added a figure, illustrating our message passing scheme, in the appendix. Please let us know if you think that it still requires a revision.

2.  We have added a discussion on the assumption of GNN generalizing across graph size (Line#122 in the main paper and Line#588 in the appendix).

3. We have also added a discussion on an alternate encoding scheme for creating message passing graphs from a given CSP (footnote on page 4 and  line#680 in appendix).

4. In the experiments section, we have added a short note on comparison with state-of-the-art and neural SAT solvers (footnote on page 7 and line#701 in the appendix.

5. Table 9 in the appendix compares the training time of different methods, followed by a discussion on the comparison (line#815).

Regards,

---

### Decision · Program_Chairs · 2022-01-20

**Decision:**

Accept (Poster)

**Comment:**

The reviewers were split about this paper: on one hand they would have liked to see more experiments on different problem settings on the other they appreciated the elegance of graph encoding methods and current results. After going through the paper and discussion I have voted to accept for the following reason:  the additional experiments and discussion posted during the rebuttal phase have addressed many of the main concerns of the reviewers (i.e., training time, message passing figure, discussion on encoding and SAT solvers). The only remaining one I see is the request for additional experiments which I don't think is grounds for rejection: current results are comprehensive and an additional experiment I think would not alter the main conclusions. I urge the authors to take all of the reviewers changes into account (if not already done so).